# Regularized Neural Ensemblers

**Sebastian Pineda Arango**[1,*]  **Maciej Janowski**[1,*]  **Lennart Purucker**[1]  **Arber Zela**[1]
**Frank Hutter**[2,3,1]  **Josif Grabocka**[4]

[1]University of Freiburg
[2]PriorLabs
[3]ELLIS Institute Tübingen
[4]University of Technology Nürnberg

**Abstract**  Ensemble methods are known for enhancing the accuracy and robustness of machine learning models by combining multiple base learners. However, standard approaches like greedy or random ensembling often fall short, as they assume a constant weight across samples for the ensemble members. This can limit expressiveness and hinder performance when aggregating the ensemble predictions. In this study, we explore employing regularized neural networks as ensemble methods, emphasizing the significance of dynamic ensembling to leverage diverse model predictions adaptively. Motivated by the risk of learning low-diversity ensembles, we propose regularizing the ensembling model by randomly dropping base model predictions during the training. We demonstrate this approach provides lower bounds for the diversity within the ensemble, reducing overfitting and improving generalization capabilities. Our experiments showcase that the regularized neural ensemblers yield competitive results compared to strong baselines across several modalities such as computer vision, natural language processing, and tabular data.

## 1 Introduction

Ensembling machine learning models is a well-established practice among practitioners and researchers, primarily due to its enhanced generalization performance over single-model predictions (Ganaie et al., 2022; Erickson et al., 2020; Feurer et al., 2015; Wang et al., 2020). Ensembles are favored for their superior accuracy and ability to provide calibrated uncertainty estimates and increased robustness against covariate shifts (Lakshminarayanan et al., 2017). Combined with their relative simplicity, these properties make ensembling the method of choice for many applications, such as medical imaging and autonomous driving, where reliability is paramount. A popular set-up consists of ensembling from a pool, after training them separately, a.k.a. post-hoc ensembling (Purucker and Beel, 2023b). This allows users to use models trained during hyperparameter optimization.

Despite these advantages, selecting post-hoc models that are accurate and diverse remains a challenging combinatorial problem, especially as the pool of candidate models grows. Commonly used heuristics, particularly in the context of tabular data, such as greedy selection (Caruana et al., 2004) and various weighting schemes, attempt to optimize ensemble performance based on metrics evaluated on a held-out validation set or through cross-validation. However, these methods face significant limitations. Specifically, choosing models to include in the ensemble and determining optimal ensembling strategies (e.g., stacking weights) are critical decisions that, if not carefully managed, can lead to overfitting on the validation data. Although neural networks are good candidates for generating ensembling weights, few studies rely on them as a post-hoc ensembling approach. We believe this happens primarily due to a lack of ensemble-related inductive biases that provide regularization.

---

*Equal contribution.

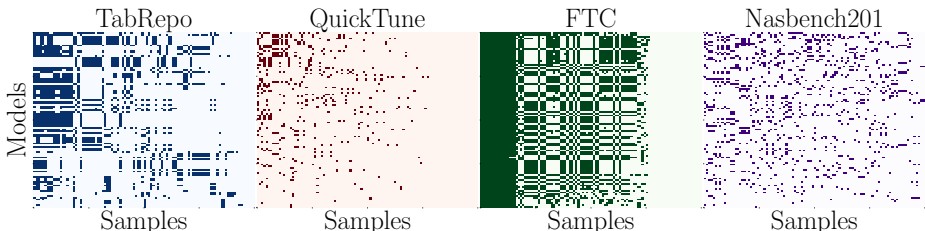

Figure 1: **Wrong Models Per Samples Across Meta-Datasets**. Every dark cell represents data instances where a model's prediction is wrong. Different models fail on different instances, therefore, only instance-specific dynamic ensembles are optimal.

In this work, we introduce a novel approach to post-hoc ensembling using neural networks. Our proposed *Neural Ensembler* dynamically generates the weights for each base model in the ensemble on a per-instance basis, a.k.a dynamical ensemble selection (Ko et al., 2008; Cavalin et al., 2013). To mitigate the risk of overfitting the validation set, we introduce a regularization technique inspired by the inductive biases inherent to the ensembling task. Specifically, we propose randomly dropping base models during training, inspired by previous work on DropOut in Deep Learning (Srivastava et al., 2014). Furthermore, our method is modality-agnostic, as it only relies on the base model predictions. For instance, for classification, we use the class predictions of the base models as input.

In summary, our contributions are as follows:

1. We propose a simple yet effective post-hoc ensembling method based on a regularized neural network that dynamically ensembles base models and is modality-agnostic.

2. To prevent the formation of low-diversity ensembles, the regularization technique randomly drops base model predictions during training. We demonstrate theoretically that this lower bounds the diversity of the generated ensemble, and validate its effect empirically.

3. Through extensive experiments, we show that Neural Ensemblers consistently select competitive ensembles across a wide range of data modalities, including tabular data (for both classification and regression), computer vision, and natural language processing.

To promote reproducibility, we have made our code publicly available in the following anonymous repository[1]. We hope that our codebase, along with the diverse set of benchmarks used in our experiments, will serve as a valuable resource for the development and evaluation of future post-hoc ensembling methods.

## 2 Background and Motivation

Post-hoc ensembling uses set of fitted base models $\{z_1, ..., z_M\}$ such that every model outputs predictions $z_m(x) : \mathbb{R}^D \to \mathbb{R}$. These outputs are combined by a stacking ensembler $f(z(x); \theta) := f(z_1(x), ..., z_M(x); \theta) : \mathbb{R}^M \to \mathbb{R}$, where $z(x) = [z_1(x), ..., z_M(x)]$ is the concatenation of the base models predictions. While the base models are fitted using a training set $\mathcal{D}_{\text{Train}}$, the ensembler's parameters $\theta$ are typically obtained by minimizing a loss function on a validation set $\mathcal{D}_{\text{Val}}$ such that:

$$\theta \in \arg\min_{\theta} \sum_{(x,y) \in \mathcal{D}_{\text{Val}}} \mathcal{L}(f(z(x); \theta), y). \tag{1}$$

In the general case, this objective function can be optimized using gradient-free optimization methods such as evolutionary algorithms (Purucker and Beel, 2023b) or greedy search (Caruana

---

[1] https://github.com/machinelearningnuremberg/RegularizedNeuralEnsemblers

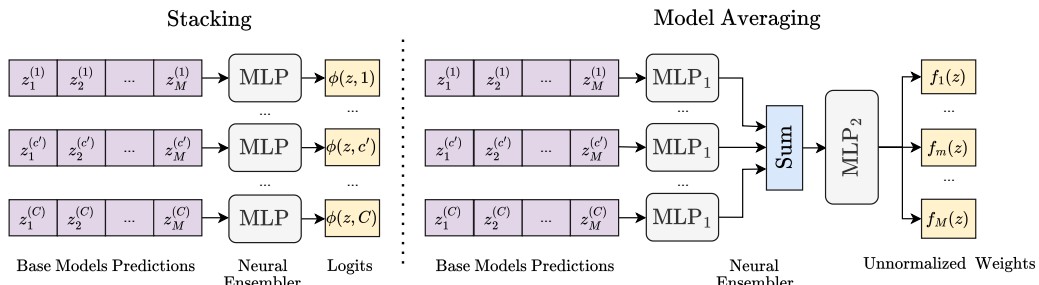

Figure 2: **Architecture of Neural Ensemblers (classification)**. The stacking mode uses a single MLP shared across base model class predictions. It outputs the logit per class, used for computing the final probability via SoftMax. In Model Averaging mode, it generates the unnormalized weights for every model, which are normalized with SoftMax.

et al., 2004). Commonly, $f$ is a linear combination $\theta \in \mathbb{R}^M$ of the model outputs:

$$f(z(x); \theta) = \sum_m \theta_m z_m(x). \tag{2}$$

Additionally, if we constraint the ensembler weights such that $\forall_i \theta_i \in \mathbb{R}_+$ and $\sum_i \theta_i = 1$ and assume probabilistic base models $z_m(x) = p(y|x, m)$, then we can interpret Equation 2 as:

$$p(y|x) = \sum_i p(y|x, m)p(m), \tag{3}$$

which is referred to as Bayesian Model Average (Raftery et al., 1997), and uses $\theta_m = p(m)$. In the general case, the probabilistic ensembler $p(y|x) = p(y|z_1(x), ..., z_M(x), \beta)$ is a stacking model parametrized by $\beta$.

## 2.1 Motivating Dynamic Ensembling

We motivate in this Section the need for dynamic ensembling by analyzing base models' predictions in real data taken from our experimental meta-datasets. Generally, the distribution $p(m)$ can take different forms. *Dynamic ensembling* assumes that the performance associated with an ensembler $f(z_m(x), \theta)$ is optimal if we select the optimal aggregation $\theta_m(x) = p(x|m)$ on a per-data point basis, instead of a static $\theta$. To motivate this observation, we selected four datasets from different modalities: *TabRepo* (Tabular data (Salinas and Erickson, 2023)), *QuickTune* (Computer Vision (Arango et al., 2024a)), *FTC* (Arango et al., 2024b) and *NasBench 201* (NAS for Computer Vision (Dong and Yang, 2020)). Then, we compute the per-sample error for 100 models in 100 samples. We report the results in Figure 1, indicating failed predictions with dark colors. We observe that models make different errors across samples, indicating a lack of optimality in static ensembling weights.

## 3 Neural Ensemblers

We build the *Neural Ensemblers* by taking the predictions of $M$ base models $z(x) = [z_1(x), ..., z_M(x)]$ as input. For regression, $z(x) : \mathbb{R}^D \to \mathbb{R}^M$ outputs the base models' point predictions given by $x \in \mathbb{R}^D$, while for classification $z^{(c)}(x) : \mathbb{R}^D \to [0, 1]^M$ returns the probabilities predicted by the base models for class $c^2$. In our discussion, we consider two functional modes for the ensemblers: as a network outputting weights for model averaging, or as a stacking model that directly outputs the

---

[2]To simplify notation, we will henceforth make the dependency implicit, denoting $z(x)$ just as $z$.

prediction. In **stacking** mode for regression, we aggregate the base model point predictions using a neural network to estimate the final prediction $\phi(z; \beta)$, where $\beta$ are the network parameters. In the **model-averaging** mode, we use a neural network to compute the weights $\theta_m(z; \beta)$ to combine the model predictions as in Equation 4.

$$\sum_m \theta_m(z; \beta) \cdot z_m \tag{4}$$

Regardless of the functional mode, the Neural Ensembler has a different output $\hat{y}$ for regression and classification. In regression, the output $\hat{y}$ is a point estimation of the mean for a normal distribution such that $p(y|x; \beta) = \mathcal{N}(\hat{y}, \sigma)$. For classification, the input is the probabilistic prediction of the base models per class $z_m^{(c)} = p(y = c|x, m)$, while the output is a categorical distribution $\hat{y} = p(y = c|z, \beta)$. We optimize $\beta$ by minimizing the negative log-likelihood over the validation dataset $\mathcal{D}_{\text{Val}}$ as:

$$\min_\beta \mathcal{L}(\beta; D_{\text{Val}}) = \min_\beta \sum_{(x,y) \in \mathcal{D}_{\text{Val}}} -\log p(y|x; \beta) \tag{5}$$

## 3.1 An Architecture with Parameter Sharing

We discuss the architectural implementation of the Neural Ensembler for the classification case, which we show in Figure 2. For the **stacking** mode, we use an MLP that receives as input the base models' predictions $z^{(c)}$ for the class $c$ and outputs the corresponding predicted logit for this class, i.e. $\phi(z, c; \beta)$. The model predictions per class are fed independently into the same network, enabling the sharing of the network parameters $\beta$ as shown in Figure 6. Subsequently, we compute the probability $p(y = c|x) = \frac{\exp^{\phi(z,c;\beta)}}{\sum_{c'} \exp^{\phi(z,c';\beta)}}$, with $\phi(x, c; \beta) = \text{MLP}(z^{(c)}; \beta)$. In regression, the final prediction is the output $\phi(z; \beta)$.

For **model averaging** mode, we use a novel architecture based on a Deep Set (Zaheer et al., 2017) embedding of the base models predictions. We compute the dynamic weights $\theta_m(z; \beta) = \frac{\exp f_m(z;\beta)}{\sum_{m'} f_{m'}(z;\beta)}$, where the unnormalized weight per model $f_m(z; \beta)$ is determined via two MLPs. The first one $\text{MLP}_1 : \mathbb{R}^M \to \mathbb{R}^H$ embeds the predictions per class $z^{(c')}$ into a latent dimension of size $H$, whereas the second network $\text{MLP}_2 : \mathbb{R}^H \to \mathbb{R}^M$ aggregates the embeddings and outputs the unnormalized weights, as shown in Equation 6. Notice that the Neural Ensemblers' input dimension and number of parameters are independent of the number of classes, due to our proposed parameter-sharing mechanism. In Appendix $C$, we further discuss the advantages of this approach from the perspective of space complexity.

$$f_m(z; \beta) = \text{MLP}_2 \left( \sum_{c'} \text{MLP}_1 \left( z^{(c')}; \beta_1 \right); \beta_2 \right) \tag{6}$$

## 3.2 Ensemble Diversity

An important aspect when building ensembles is guaranteeing diversity among the models (Wood et al., 2023; Jeffares et al., 2024). This has motivated some approaches to explicitly account for diversity when searching the ensemble configuration (Shen et al., 2022; Purucker et al., 2023). A common way to measure diversity is the *ambiguity* (Krogh and Vedelsby, 1994), which can be derived after decomposing the loss function (Jeffares et al., 2024). Measuring diversity is essential as it helps to avoid overfitting in the ensemble.

**Definition 3.1** (Ensemble Diversity via Ambiguity Measure). An ensemble with base models $\mathcal{Z} = z_1, ..., z_M$ with prediction $\bar{z} = \sum_m \theta_m z_m$ has diversity $\alpha := \mathbb{E} \left[ \sum_m \theta_m (z_m - \bar{z})^2 \right]$.

**Algorithm 1:** Training Algorithm for Neural Ensemblers with Base Models' DropOut

---

**Input**: Base model predictions $\{z_1(x), ..., z_M(x)\}$, validation data $\mathcal{D}_{\text{Val}}$, probability of retaining $\gamma$, $mode \in \{\text{Stacking, Averaging}\}$.

**Output**: Neural Ensembler's parameters $\beta$

1 Initialize randomly parameters $\beta$ ;
2 **while** *done* **do**
3     Sample masking vector $r \in \mathbb{R}^M, r_i \sim \text{Ber}(\gamma)$;
4     Mask base models predictions $z_{\text{drop}} = r \odot z$ ;
5     **if** *mode* = Stacking **then**
6        Compute predictions $\phi\left(\frac{1}{\gamma}z_{\text{drop}}\right)$ ;
7     **else**
8        Compute weights $\theta_m\left(\frac{1}{\gamma}z_{\text{drop}}\right)$ using Equation 7 ;
9        Compute predictions using Equation 4 ;
10     **end**
11     Update parameters $\beta$ using $\nabla \mathcal{L}(\beta; \mathcal{D}_{\text{Val}})$
12 **end**
13 **return** $\beta$;

---

### 3.3 Base Models' DropOut

Using Neural Ensemblers tackles the need for dynamical ensembling. Moreover, it gives additional expressivity associated with neural networks. However, there is also a risk of overfitting caused by a low diversity. Although the error might effectively decrease the validation loss (Equation 5), it does not necessarily generalize to test samples. Inspired by previous work (Srivastava et al., 2014), we propose to drop some base models during training forward passes. Intuitively, this forces the ensembler to rely on different base models to perform the predictions, instead of merely using the preferred base model(s).

Formally, we mask the inputs such as $r_m \cdot z_m$, where $r_m \sim \text{Ber}(\gamma)$, where $\text{Ber}(\gamma)$ is the Bernoulli distribution with parameter $\gamma$ with represents the probability of keeping the base model, while $\delta = 1 - \gamma$ represents the DropOut rate. We also mask the weights when using model averaging:

$$\theta_m(z; \beta, r) = \frac{r_m \cdot \exp f_m(z; \beta)}{\sum_{m'} r_{m'} \cdot \exp f_{m'}(z; \beta)}. \tag{7}$$

As *DropOut* changes the scale of the inputs, we should apply the *weight scaling rule* during inference by multiplying the dropped variables by the retention probability $\gamma$. Alternatively, we can scale the variables during training by $\frac{1}{\gamma}$. In Algorithm 1, we detail how to train the Neural Ensemblers by dropping base model predictions. It has two modes, acting as a direct stacker or as a model averaging ensembler. We demonstrate that base models' DropOut lower bounds the diversity for a simple ensembling case in Proposition A.1.

**Definition 3.2 (Preferred Base Model).** Consider a target variable $y \in R$ and a set of uncorrelated base models predictions $\mathcal{Z} = \{z_m | z_m \in \text{R}, m = 1, ..., M\}$. $z_p$ is the *Preferred Base Model* if it has the highest *sample* correlation to the target, i.e. $\rho_{z_p,y} \in [0, 1], \rho_{z_p,y} > \rho_{z_m,y}, \forall z_m \in \mathcal{Z}/\{z_p\}$.

**Proposition 3.3 (Diversity Lower Bound).** *As the correlation of the preferred model increases $\rho_{p_m,y} \rightarrow 1$, the diversity decreases $\alpha \rightarrow 1 - \gamma$, when using Base Models' DropOut with probability of retaining $\gamma$.*

Table 1: Average Normalized NLL across Metadatasets.

| | FTC | NB-Micro | NB-Mini | QT-Micro | QT-Mini | TR-Class | TR-Reg |
|---|---|---|---|---|---|---|---|
| Single-Best | $1.0000_{\pm0.0000}$ | $1.0000_{\pm0.0000}$ | $1.0000_{\pm0.0000}$ | $1.0000_{\pm0.0000}$ | $1.0000_{\pm0.0000}$ | $1.0000_{\pm0.0000}$ | $1.0000_{\pm0.0000}$ |
| Random | $1.5450_{\pm0.5289}$ | $0.6591_{\pm0.2480}$ | $0.7570_{\pm0.2900}$ | $6.8911_{\pm3.1781}$ | $5.8577_{\pm3.2546}$ | $1.7225_{\pm1.9645}$ | $1.8319_{\pm2.1395}$ |
| Top5 | $0.8406_{\pm0.0723}$ | $0.6659_{\pm0.1726}$ | $0.6789_{\pm0.3049}$ | $1.5449_{\pm1.8358}$ | $1.1496_{\pm0.3684}$ | $1.0307_{\pm0.5732}$ | $\underline{0.9939}_{\pm0.0517}$ |
| Top50 | $0.8250_{\pm0.1139}$ | $0.5849_{\pm0.2039}$ | $0.6487_{\pm0.3152}$ | $3.3068_{\pm2.6197}$ | $3.0618_{\pm2.2960}$ | $1.0929_{\pm1.0198}$ | $1.0327_{\pm0.2032}$ |
| Quick | $0.7273_{\pm0.0765}$ | $0.5957_{\pm0.1940}$ | $0.6497_{\pm0.3030}$ | $1.1976_{\pm1.1032}$ | $0.9747_{\pm0.2082}$ | $\underline{0.9860}_{\pm0.2201}$ | $1.0211_{\pm0.1405}$ |
| Greedy | $\underline{0.6943}_{\pm0.0732}$ | $\underline{0.5785}_{\pm0.1972}$ | $0.7617_{\pm0.3435}$ | $\underline{0.9025}_{\pm0.2378}$ | $\underline{0.9093}_{\pm0.1017}$ | $\mathbf{0.9665}_{\pm0.0926}$ | $1.0149_{\pm0.1140}$ |
| CMAES | $1.2356_{\pm0.5295}$ | $1.0000_{\pm0.0000}$ | $1.0000_{\pm0.0000}$ | $4.1728_{\pm2.8724}$ | $4.6474_{\pm3.0180}$ | $1.3487_{\pm1.3390}$ | $1.0281_{\pm0.1977}$ |
| Random Forest | $0.7496_{\pm0.0940}$ | $0.8961_{\pm0.3159}$ | $0.9340_{\pm0.4262}$ | $3.7033_{\pm2.8145}$ | $2.2938_{\pm2.2068}$ | $1.2655_{\pm0.4692}$ | $1.0030_{\pm0.0871}$ |
| Gradient Boosting | $0.7159_{\pm0.1529}$ | $1.7288_{\pm1.2623}$ | $1.2575_{\pm0.4460}$ | $1.9373_{\pm1.2839}$ | $2.6193_{\pm2.3159}$ | $1.4288_{\pm1.2083}$ | $1.0498_{\pm0.2128}$ |
| SVM | $0.7990_{\pm0.0909}$ | $0.7744_{\pm0.2967}$ | $0.9358_{\pm0.5706}$ | $5.4377_{\pm3.3807}$ | $4.0019_{\pm3.6601}$ | $1.3884_{\pm1.4276}$ | $2.7975_{\pm3.0219}$ |
| Linear | $0.7555_{\pm0.0898}$ | $0.7400_{\pm0.2827}$ | $0.8071_{\pm0.2206}$ | $1.3960_{\pm1.2334}$ | $1.1031_{\pm0.7038}$ | $1.1976_{\pm1.1024}$ | $3.1488_{\pm3.2813}$ |
| XGBoost | $0.8292_{\pm0.1434}$ | $0.7389_{\pm0.2326}$ | $0.9092_{\pm0.5304}$ | $3.7822_{\pm3.1194}$ | $2.6119_{\pm2.3911}$ | $1.7697_{\pm1.4672}$ | $1.2580_{\pm0.4875}$ |
| CatBoost | $\mathbf{0.6887}_{\pm0.0953}$ | $0.8092_{\pm0.2513}$ | $0.9512_{\pm0.5083}$ | $2.6262_{\pm2.6482}$ | $2.4145_{\pm1.8989}$ | $1.2570_{\pm1.2859}$ | $1.0454_{\pm0.1550}$ |
| LightGBM | $0.7973_{\pm0.1946}$ | $3.6004_{\pm2.5822}$ | $5.3943_{\pm4.7980}$ | $3.0378_{\pm2.7945}$ | $3.6860_{\pm3.2856}$ | $1.8298_{\pm1.1596}$ | $1.6250_{\pm2.1651}$ |
| Akaike | $0.8526_{\pm0.1403}$ | $0.5838_{\pm0.2031}$ | $\underline{0.6485}_{\pm0.3166}$ | $3.1574_{\pm2.5898}$ | $2.6888_{\pm2.0620}$ | $1.0930_{\pm1.0203}$ | $1.0221_{\pm0.1793}$ |
| MA | $0.9067_{\pm0.1809}$ | $0.5970_{\pm0.2034}$ | $0.6530_{\pm0.3028}$ | $4.7921_{\pm3.0780}$ | $4.0168_{\pm2.8560}$ | $1.4724_{\pm1.9401}$ | $1.3342_{\pm1.3515}$ |
| DivBO | $0.7695_{\pm0.1195}$ | $0.7307_{\pm0.3061}$ | $0.6628_{\pm0.3435}$ | $1.2251_{\pm1.0293}$ | $0.9430_{\pm0.2036}$ | $1.0023_{\pm0.3411}$ | $1.0247_{\pm0.1473}$ |
| EO | $0.7535_{\pm0.1156}$ | $0.5801_{\pm0.2051}$ | $0.6911_{\pm0.2875}$ | $1.3702_{\pm1.6389}$ | $0.9649_{\pm0.2980}$ | $1.0979_{\pm1.1289}$ | $1.0183_{\pm0.0993}$ |
| NE-Stack | $0.7562_{\pm0.1836}$ | $\mathbf{0.5278}_{\pm0.2127}$ | $\mathbf{0.6336}_{\pm0.3456}$ | $\mathbf{0.7486}_{\pm0.6831}$ | $\mathbf{0.6769}_{\pm0.2612}$ | $1.3268_{\pm0.7498}$ | $1.2379_{\pm0.4083}$ |
| NE-MA | $0.6952_{\pm0.0730}$ | $0.5822_{\pm0.2147}$ | $0.6522_{\pm0.3131}$ | $1.0177_{\pm0.5151}$ | $0.9166_{\pm0.0936}$ | $1.0515_{\pm1.0003}$ | $\mathbf{0.9579}_{\pm0.0777}$ |

*Sketch of Proof.* We want to compute $\lim_{\rho_{z_p,y}\to1} \alpha = \lim_{\rho_{z_p,y}\to1} \mathbb{E}\left[\sum_m \theta_m(z_m - \bar{z})^2\right]$. By using $\bar{z} = \sum_m r_m \, \theta_m z_m$, and assuming, without loss of generality, that the predictions are standardized, we obtain $\mathbb{V}(r \cdot z_m) = \gamma$. This lead as to $\lim_{\rho_{z_p,y}\to1} \alpha = 1 - \gamma$, after following a procedure similar to Proposition 3.1. We provide the complete proof in Appendix A.

## 4 Experiments and Results

In this section, we provide empirical evidence of the effectiveness of our approach.

### 4.1 Experimental Setup

**Meta-Datasets.**. In our experiments, we utilize four meta-datasets with pre-computed predictions, which allows us to simulate ensembles without the need to fit models. These meta-datasets cover diverse data modalities, including Computer Vision, Tabular Data, and Natural Language Processing. Additionally, we evaluate the method on datasets without pre-computed predictions to assess the performance of ensembling methods that rely on fitted models during the evaluation. Table 11 reports the main information related to these datasets. Particularly for *Nasbench*, we created 2 versions by subsampling 100 and 1000 models. The meta-dataset *Finetuning Text Classifiers (FTC)* (Arango et al., 2024b) contains the predictions of several language models to evaluate ensembling techniques on text classification tasks by finetuning language models such as GPT2 (Radford et al., 2019), Bert (Devlin et al., 2018) and Bart (Lewis et al., 2019). We also generate a set of fitted *Scikit-Learn Pipelines* on classification datasets. In this case, we stored the pipeline in memory, allowing us to evaluate our method in practical scenarios where the user has fitted models instead of predictions. We detailed information about the creation of this meta-dataset in Appendix O. Altogether, the meta-datasets comprise over 240 datasets in total. The information about each data set lies in the referred work under the column *Task Information* in Table 11.

**Baselines.**. We compare the **Neural Ensemblers (NE)** with other common and competitive ensemble approaches. 1) **Single best** selects the best model according to the validation metric; 2) **Random** chooses randomly $N = 50$ models to ensemble, 3) **Top-N** ensembles the best $N$ models according to the validation metric; 4) **Greedy** creates an ensemble with $N = 50$ models by iterative selecting the

one that improves the metric as proposed by previous work (Caruana et al., 2004); 5) **Quick** builds the ensemble with 50 models by adding model subsequently only if they strictly improve the metric; 6) **CMAES** (Purucker and Beel, 2023b) uses an evolutive strategy with a post-processing method for ensembling, 7) **Model Average (MA)** computed the sum of the predictions with constant weights as in Equation 3. We also compare to methods that perform ensemble search iteratively via Bayesian Optimization such as 8) **DivBO** (Shen et al., 2022), and 9) **Ensemble Optimization (EO)** (Levesque et al., 2016). We also report results by using common ML models as stackers, such as **SVM** and **Random Forest**. Finally, we include models from Dynamic Ensemble Search (DES) literature such as **KNOP** (Cavalin et al., 2013), **KNORAE** (Ko et al., 2008) and **MetaDES** (Cruz et al., 2015). We provide further details on the baselines setup in Appendix E.

**Neural Ensemblers' Setup.**. We train the neural networks for 10000 update steps, with a batch size of 2048. If the GPU memory is not enough for fitting the network because of the number of base models, or the number of classes, we decreased the batch size to 256. Additionally, we used the Adam optimizer and a network with four layers, 32 neurons, and a probability of keeping base models $\gamma = 0.25$, or alternatively a DropOut rate $\delta = 0.75$. Notice that the architecture of the ensemblers slightly varies depending on the mode, *Stacking* or *Model Average* (MA). For the Stacking mode, we use an MLP with four layers and 32 neurons with ReLU activations. For MA mode, we use two MLPs as in Equation 6: 1) $MLP_1$ has 3 layers with 32 neurons, while 2) $MLP_2$ has one layer with the same number of neurons. Although changing some of these hyperparameters might improve the performance, we keep this setup constant for all the experiments, after checking that the Neural Ensemblers perform well in a subset of the Quick-Tune meta-dataset (*extended version*).

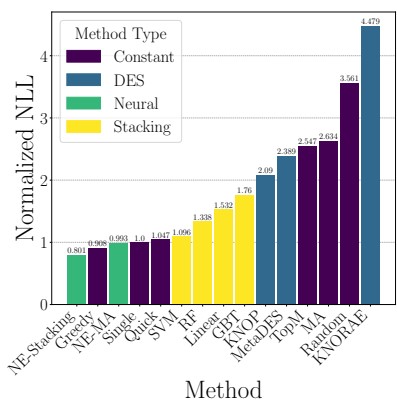

Figure 3: Results on Scikit-Learn Pipelines.

### 4.2 Research Questions and Associated Experiments

**RQ 1**: **Can Neural Ensemblers outperform other common and competitive ensembling methods across data modalities?**

   **Experimental Protocol**. To answer this question, we compare the neural ensembles in stacking and averaging mode to the baselines across all the meta-datasets. We run every ensembling method three times for every dataset. In all the methods we use the validation data for fitting the ensemble, while we report the results on the test split. Specifically, we report the average across datasets of two metrics: negative log-likelihood (NLL) and classification error. For the tabular classification, we compute the ROC-AUC. As these metrics vary for every dataset, we normalize metrics by dividing them by the *single-best* metric. Therefore, a method with a normalized metric below one is improving on top of using the single best base model. We report the standard deviation across the experiments per dataset and highlight in bold the best method.

   **Results**. The results reported in Table 2 and Table 1 show that our proposed regularized neural networks **are competitive post-hoc ensemblers**. In general, we observe that the Neural Ensemblers variants obtain either the best (in bold) or second best (italic) performance across almost all meta-datasets and metrics. Noteworthily, the greedy approach is very competitive, especially for the *FTC* and *TR-Class* meta-datasets. This is coherent with previous work supporting greedy ensembling as a robust method for tabular data (Erickson et al., 2020). We hypothesize that dynamic ensembling

Table 2: Average Normalized Error across Metadatasets.

| | FTC | NB-Micro | NB-Mini | QT-Micro | QT-Mini | TR-Class | TR-Class (AUC) |
|---|---|---|---|---|---|---|---|
| Single-Best | $1.0000_{\pm 0.0000}$ | $1.0000_{\pm 0.0000}$ | $1.0000_{\pm 0.0000}$ | $1.0000_{\pm 0.0000}$ | $1.0000_{\pm 0.0000}$ | $1.0000_{\pm 0.0000}$ | $1.0000_{\pm 0.0000}$ |
| Random | $1.3377_{\pm 0.2771}$ | $0.7283_{\pm 0.0752}$ | $0.7491_{\pm 0.2480}$ | $6.6791_{\pm 3.4638}$ | $4.7284_{\pm 2.9463}$ | $1.4917_{\pm 1.6980}$ | $1.7301_{\pm 1.8127}$ |
| Top5 | $0.9511_{\pm 0.0364}$ | $0.6979_{\pm 0.0375}$ | $0.6296_{\pm 0.1382}$ | $\underline{0.6828}_{\pm 0.3450}$ | $0.8030_{\pm 0.2909}$ | $0.9998_{\pm 0.1233}$ | $0.9271_{\pm 0.2160}$ |
| Top50 | $1.1012_{\pm 0.1722}$ | $0.6347_{\pm 0.0395}$ | $0.5650_{\pm 0.1587}$ | $1.0662_{\pm 0.9342}$ | $1.0721_{\pm 0.4671}$ | $0.9800_{\pm 0.1773}$ | $0.9297_{\pm 0.2272}$ |
| Quick | $0.9494_{\pm 0.0371}$ | $0.6524_{\pm 0.0436}$ | $0.5787_{\pm 0.1510}$ | $0.7575_{\pm 0.2924}$ | $\underline{0.7879}_{\pm 0.2623}$ | $0.9869_{\pm 0.1667}$ | $\underline{0.9054}_{\pm 0.2232}$ |
| Greedy | $0.9494_{\pm 0.0374}$ | $0.7400_{\pm 0.1131}$ | $0.6033_{\pm 0.1572}$ | $0.9863_{\pm 0.4286}$ | $0.9297_{\pm 0.1435}$ | $0.9891_{\pm 0.1693}$ | $0.9090_{\pm 0.2197}$ |
| CMAES | $\underline{0.9489}_{\pm 0.0392}$ | $0.6401_{\pm 0.0343}$ | $0.5797_{\pm 0.1575}$ | $1.0319_{\pm 0.5000}$ | $0.9086_{\pm 0.1121}$ | $0.9935_{\pm 0.1953}$ | $1.1878_{\pm 1.1457}$ |
| Random Forest | $0.9513_{\pm 0.0359}$ | $0.6649_{\pm 0.0427}$ | $0.6891_{\pm 0.3039}$ | $1.4738_{\pm 1.3510}$ | $1.2530_{\pm 0.4875}$ | $1.0041_{\pm 0.2330}$ | $1.0924_{\pm 0.6284}$ |
| Gradient Boosting | $1.0097_{\pm 0.1033}$ | $1.2941_{\pm 0.5094}$ | $1.2037_{\pm 0.3528}$ | $0.8514_{\pm 0.5003}$ | $1.6121_{\pm 1.7023}$ | $1.0452_{\pm 0.3808}$ | $1.0663_{\pm 0.4884}$ |
| SVM | $\mathbf{0.9453}_{\pm 0.0383}$ | $0.6571_{\pm 0.0483}$ | $0.7015_{\pm 0.3067}$ | $1.1921_{\pm 0.8266}$ | $1.4579_{\pm 0.6233}$ | $\mathbf{0.9585}_{\pm 0.2160}$ | $1.4701_{\pm 1.3486}$ |
| Linear | $0.9609_{\pm 0.0347}$ | $0.7891_{\pm 0.1978}$ | $0.7782_{\pm 0.1941}$ | $0.7333_{\pm 0.4457}$ | $0.9291_{\pm 0.3580}$ | $0.9776_{\pm 0.2844}$ | $1.0329_{\pm 0.4022}$ |
| XGBoost | $0.9784_{\pm 0.0334}$ | $0.7886_{\pm 0.0903}$ | $0.7183_{\pm 0.2628}$ | $3.3875_{\pm 3.1419}$ | $2.1067_{\pm 1.9570}$ | $1.3310_{\pm 1.4402}$ | $1.1490_{\pm 1.0202}$ |
| CatBoost | $1.0271_{\pm 0.0952}$ | $0.9247_{\pm 0.0652}$ | $0.8526_{\pm 0.2627}$ | $1.1858_{\pm 0.5737}$ | $1.4011_{\pm 0.7817}$ | $1.1507_{\pm 1.0190}$ | $1.0179_{\pm 0.2395}$ |
| LightGBM | $0.9774_{\pm 0.0369}$ | $1.2866_{\pm 0.6738}$ | $1.2278_{\pm 0.8503}$ | $2.4846_{\pm 2.5747}$ | $3.1337_{\pm 2.9066}$ | $1.0954_{\pm 0.4326}$ | $1.1228_{\pm 0.9022}$ |
| Akaike | $1.0242_{\pm 0.0723}$ | $\underline{0.6328}_{\pm 0.0384}$ | $0.5667_{\pm 0.1578}$ | $1.0323_{\pm 0.8817}$ | $1.0346_{\pm 0.4554}$ | $0.9832_{\pm 0.1784}$ | $1.1146_{\pm 1.0218}$ |
| MA | $1.0709_{\pm 0.0845}$ | $0.6381_{\pm 0.0349}$ | $\mathbf{0.5610}_{\pm 0.1490}$ | $1.1548_{\pm 0.8465}$ | $1.2173_{\pm 0.6107}$ | $1.0917_{\pm 1.0135}$ | $0.9977_{\pm 0.2278}$ |
| DivBO | $1.0155_{\pm 0.1452}$ | $0.6915_{\pm 0.0536}$ | $0.9120_{\pm 0.1524}$ | $1.3935_{\pm 1.4316}$ | $1.0635_{\pm 0.7587}$ | $1.0908_{\pm 1.0104}$ | $1.0899_{\pm 1.0297}$ |
| EO | $1.0208_{\pm 0.1159}$ | $0.6365_{\pm 0.0445}$ | $0.5704_{\pm 0.1619}$ | $1.0185_{\pm 0.6464}$ | $1.0367_{\pm 0.4394}$ | $1.0851_{\pm 1.0136}$ | $0.9377_{\pm 0.2310}$ |
| NE-Stack | $0.9491_{\pm 0.0451}$ | $0.6331_{\pm 0.0378}$ | $0.5836_{\pm 0.1592}$ | $\mathbf{0.6104}_{\pm 0.3656}$ | $\mathbf{0.7545}_{\pm 0.2960}$ | $1.0440_{\pm 0.3309}$ | $1.0035_{\pm 0.5295}$ |
| NE-MA | $0.9527_{\pm 0.0402}$ | $\mathbf{0.6307}_{\pm 0.0363}$ | $\underline{0.5621}_{\pm 0.1483}$ | $0.8297_{\pm 0.4974}$ | $0.8236_{\pm 0.2240}$ | $\underline{0.9592}_{\pm 0.2144}$ | $\mathbf{0.9028}_{\pm 0.2157}$ |

contributes partially to the strong results for the Neural Ensemblers. However, the expressivity gained is not enough, because it can lead to overfitting. To understand this, we compare to Dynamic Ensemble Selection (DES) methods. Specifically, we use *KNOP*, *MetaDES*, and *KNORAE*, and evaluate all methods in *Scikit-learn Pipelines* meta-dataset, as we can easily access the fitted models. We report the results of the test split in Figure 3, where we distinguish among four types of models to facilitate the reading: *Neural, DES, Stacking* and *Constant*. We can see that Neural Ensemblers are the most competitive approaches, especially on *stacking* mode. Additionally, we report the metrics on the validation split in Figure 6 (Appendix F), where we observe that some dynamic ensemble approaches such as *Gradient Boosting (GBT)*, *Random Forest (RF)* and *KNORAE* exhibit overfitting, while Neural Enemblers are more robust.

**RQ 2: What is the impact of the DropOut regularization scheme?**

**Experimental Protocol**. To understand how much the base learners DropOut helps the Neural Ensemblers, we run an ablation by trying the following values for the DropOut rate $\delta \in \{0.0, 0.1, \ldots, 0.9\}$. We compute the average NLL for three seeds per dataset and divide this value by the one obtained for $\delta = 0.0$ in the same dataset. In this setup, we realize that a specific DropOut rate is improving over the default network without regularization if the normalized NLL is below 1.

**Results**. Our ablation study demonstrates that non-existing or high DropOut are detrimental to the Neural Ensembler performance in general. As shown in Figure 4, this behavior is consistent in all datasets and both modes, but *TR-Reg* metadataset on *Stacking* mode. In general, we observe that **Neural Ensemblers obtain better performance when using base models' DropOut**. Furthermore, we show how the mean weight per model is related to the mean error of the models for different DropOut rates in Figure 15 in the Appendix N.

**RQ 3: How sensible are the Neural Ensemblers to its hyperparameters?**

**Experimental Protocol**. While our proposed Neural Ensembler (NE) uses MLPs with 4 layers and 32 neurons per layer by default, it is important to understand how sensitive the NE's performance is to changes in its hyperparameters. To this end, we conduct an extensive ablation study by varying the number of layers $L \in \{1, 2, 3, 4, 5\}$ and the number of neurons per layer (hidden dimension) $H \in \{8, 16, 32, 64, 128\}$ in the stacking mode of the NE. For each configuration, we compute the Negative Log Likelihood (NLL) on every task in the metadatasets and normalize these values by

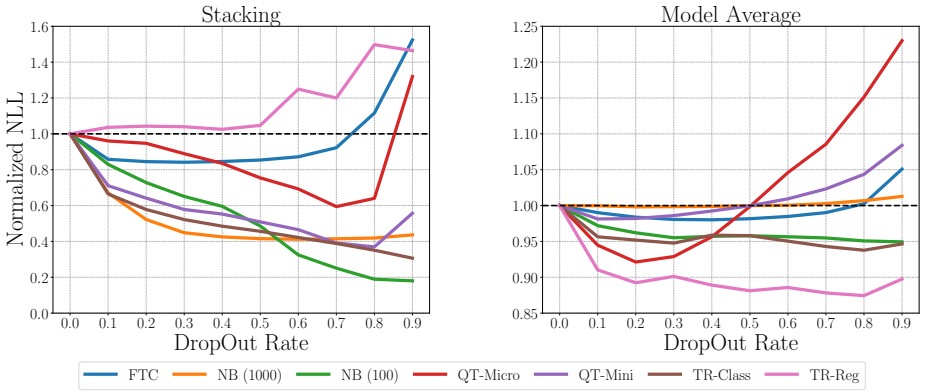

Figure 4: Ablation of the DropOut rate.

the performance obtained with the default hyperparameters ($L = 4$, $H = 32$). This normalization allows us to compare performance changes across different tasks and metadatasets, accounting for differences in metric scales. A normalized NLL value below one indicates improved performance compared to the default setting.

**Results**. Our findings indicate that there is no single hyperparameter configuration that is optimal across all datasets. However, our default configuration ($L = 4$, $H = 32$) strikes a balance, providing robust performance across diverse tasks without the need for extensive hyperparameter tuning, and can be effectively applied in various settings without the need for dataset-specific hyperparameter optimization.

**Significance on D Datasets with a lot of Classes.**. Given the high performance between *Greedy* and **NE-MA**, we wanted to understand when the second one would obtain strong significant results. We found that that **NE-MA** is particularly well-performing in datasets with a large number of classes. Given Table 11, we can see that four meta-datasets have a high (> 10) number of classes, thus they have datasets with a lot of classes. We selected these metadatasets (*NB(100), NB(1000), QT-Micro, QT-Mini*), and plotted the significance compared to *Greedy* and *Random Search*. The results reported in Figure 9 demonstrate that our approach is significantly better than Greedy in these metadatasets.

**Additional research questions.**. We further discuss interesting research questions in the Appendix due to the limited space. Some of these include: *i)* how much the validation data size affects the Neural Ensembler (Appendix J), *ii)* what happens if we train the Neural Ensemblers on a merged split containing both training and validation data (Appendix K), and *iii)* the disadvantages of Neural Ensemblers acting on the original input space (Appendix L).

## 5  Related Work

**Ensembles for Tabular Data**. For tabular data, ensembles are known to perform better than individual models (Sagi and Rokach, 2018; Salinas and Erickson, 2023).Therefore, ensembles are often used in real-world applications (Dong et al., 2020), to win competitions (Koren, 2009; Kaggle, 2024), and by automated machine learning (AutoML) systems as a modeling strategy (Purucker and Beel, 2023b; Purucker et al., 2023). Methods like Bagging (Breiman, 1996) or Boosting (Freund et al., 1996) are often used to boost the performance of individual models. In contrast, post-hoc ensembling (Shen et al., 2022; Purucker and Beel, 2023a) aggregates the predictions of an arbitrary set of fitted base models. Post-hoc ensembles can be built by stacking (Wolpert, 1992) ensemble selection (a.k.a. pruning) (Caruana et al., 2004) , or through a systematic search for an optimal ensemble (Levesque et al., 2016).

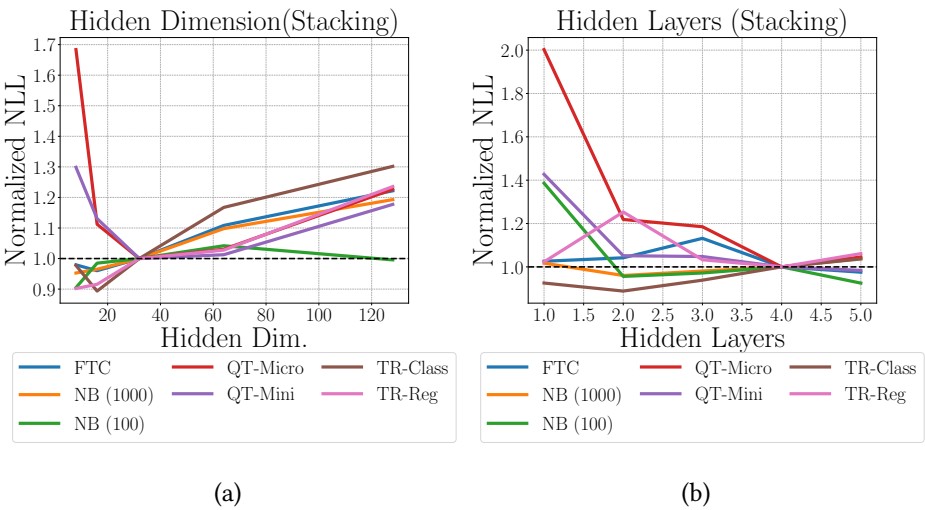

Figure 5: Ablation study of Neural Ensembler hyperparameters: number of neurons per layers (a) and number of layers (b). Normalized NLL values below one indicate improved performance over the default setting ($L = 4$, $H = 32$).

**Ensembles for Deep Learning**. Ensembles of neural networks (Hansen and Salamon, 1990; Krogh and Vedelsby, 1994; Dietterich, 2000) have gained significant attention in deep learning research, both for their performance-boosting capabilities and their effectiveness in uncertainty estimation. Various strategies for building ensembles exist, with deep ensembles (Lakshminarayanan et al., 2017) being the most popular one, which involves independently training multiple initializations of the same network. Extensive empirical studies (Ovadia et al., 2019; Gustafsson et al., 2020) have shown that deep ensembles outperform other approaches for uncertainty estimation, such as Bayesian neural networks (Blundell et al., 2015; Gal and Ghahramani, 2016; Welling and Teh, 2011).

**Dynamic Ensemble Selection**. Our Neural Ensembler is highly related to dynamic ensemble selection. Both dynamically aggregate the predictions of base models per instance (Cavalin et al., 2013; Ko et al., 2008). Traditional dynamic ensemble selection methods aggregate the most competent base models by paring heuristics to measure competence with clustering, nearest-neighbor-based, or traditional tabular algorithms (like naive Bayes) as meta-models (Cruz et al., 2018, 2020). In contrast, we use an end-to-end trained neural network to select *and weight* the base models per instance.

**Mixture-of-Experts**. Our idea of generating ensemble base model weights is closely connected to the mixture-of-experts (MoE) (Jacobs et al., 1991; Jordan and Jacobs, 1993; Shazeer et al., 2017), where one network is trained with specialized sub-modules that are activated based on the input data. Alternatively, we could include a layer, aggregating predictions by encouraging diversity (Zhang et al., 2020). In contrast to these approaches, our Neural Ensemblers can ensemble any (black-box) model and are not restricted to gradient-based approaches.

## 6 Conclusions

In this work, we tackled the challenge of post-hoc ensemble selection and the associated risk of overfitting on the validation set. We introduced the *Neural Ensembler*, a neural network that dynamically assigns weights to base models on a per-instance basis. To reduce overfitting, we proposed a regularization technique that randomly drops base models during training, which we theoretically showed enhances ensemble diversity. Our empirical results demonstrated that Neural Ensemblers consistently form competitive ensembles across diverse data modalities, including tabular data (classification and regression), computer vision, and natural language processing.

## Acknowledgments

The authors gratefully acknowledge the scientific support and HPC resources provided by the Erlangen National High Performance Computing Center (NHR@FAU) of the Friedrich-Alexander-Universität Erlangen-Nürnberg (FAU) under the NHR project v101be. NHR funding is provided by federal and Bavarian state authorities. NHR@FAU hardware is partially funded by the German Research Foundation (DFG) – 440719683. This research was partially supported by the following sources: TAILOR, a project funded by EU Horizon 2020 research and innovation programme under GA No 952215; the Deutsche Forschungsgemeinschaft (DFG, German Research Foundation) under grant number 417962828 and 499552394 - SFB 1597; the European Research Council (ERC) Consolidator Grant "Deep Learning 2.0" (grant no. 101045765). Frank Hutter acknowledges financial support by the Hector Foundation. The authors acknowledge support from ELLIS and ELIZA. Funded by the European Union. Views and opinions expressed are however those of the author(s) only and do not necessarily reflect those of the European Union or the ERC. Neither the European Union nor the ERC can be held responsible for them.

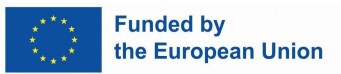

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

## Appendix

We want to summarize here all the appendix sections:

- Section A presents the proofs of propositions in the main paper.

- Section B discusses the limitations of our proposed method and its broader impacts.

- Section D further details research similar to our work.

- Section O explains the process of gathering and preparing the *FTC* and *Scikit-learn Pipelines* metadatasets used in our experiments.

- Section F includes supplementary tables and figures, such as average ranks and detailed performance metrics, that support and expand upon the main experimental results reported in the paper.

- Section H includes he computational cost associated with our method compared to baseline approaches, including runtime evaluations and discussions on efficiency.

- Section I includes the results of a proof-of-concept experiment using overparameterized base models (e.g., 10th-degree polynomials), demonstrating the effectiveness of our Neural Ensembler even when base models have high capacity.

- Section G includes the Critical Difference diagrams corresponding to our main results, illustrating the statistical significance of performance differences among methods and how to interpret these diagrams.

- Section J explores the impact of using different amounts of validation data to train the Neural Ensembler, assessing its sample efficiency and how performance scales with varying data sizes.

- Section K explores the effect of merging the training and validation datasets on the performance of both base models and ensemblers.

- Section L explores an alternative formulation of the Neural Ensembler that operates on the original input space rather than on the predictions of the base model, including experimental results and discussions of its effectiveness.

- In section M, we study whether the performance of the models changes when using base models found by Bayesian Optimization or randomly.

## A  Proofs

**Proposition A.1** (Diversity Lower Bound)*. As the correlation of the preferred model $\rho_{p_m,y} \rightarrow 1$, the diversity $\alpha \rightarrow 1 - \gamma$, when using Base Models' DropOut with probability of retaining $\gamma$.*

*Proof.* Without loss of generality, we assume that the random variables are standardized. We follow a similar procedure as for Proposition 2, by considering $\bar{z} = \sum_m r_m \, \theta_m z_m$. We demonstrate that $\mathbb{V}(r \cdot z_m) = \gamma$, given that $r \sim \text{Bernoulli}(\gamma)$.

$$\mathbb{V}(r_m \cdot z_m) = \mathbb{V}(r_m) \cdot \mathbb{V}(z_m) + \mathbb{V}(z_m) \cdot \mathbb{E}(r_m)^2 + \mathbb{V}(r_m) \cdot \mathbb{E}(z_m)^2 \tag{8}$$

$$\mathbb{V}(r_m \cdot z_m) = \mathbb{V}(r_m) + \mathbb{E}(r_m)^2 \tag{9}$$

$$\mathbb{V}(r_m \cdot z_m) = \gamma(1 - \gamma) + \gamma^2 \tag{10}$$

$$\mathbb{V}(r_m \cdot z_m) = \gamma. \tag{11}$$

Then, we evaluate the variance of the ensemble using DropOut $\mathbb{V}(\bar{z})$:

$$\mathbb{V}(\bar{z}) = \mathbb{V}\left(\sum_m r_m \cdot \theta_m \cdot z_m\right) \tag{12}$$

$$\mathbb{V}(\bar{z}) = \sum_m \mathbb{V}(r_m \cdot \theta_m \cdot z_m) \tag{13}$$

$$\mathbb{V}(\bar{z}) = \sum_m \theta_m^2 \mathbb{V}(r_m \cdot z_m) \tag{14}$$

$$\mathbb{V}(\bar{z}) = \gamma \sum_m \theta_m^2. \tag{15}$$

Applying Equation 15 into Equation ??, we obtain:

$$\lim_{\rho_{z_p,y}\to 1} \alpha = \lim_{\rho_{z_p,y}\to 1} \mathbb{E}\left[\sum_m \gamma_m \cdot \theta_m (z_m - \bar{z})^2\right] \tag{16}$$

$$= \lim_{\rho_{z_p,y}\to 1} \sum_m \theta_m \left(1 - \gamma \sum_m \theta_m^2\right) \tag{17}$$

$$= \lim_{\rho_{z_p,y}\to 1} \left(1 - \gamma \sum_{m'} \rho_{z_{m'},y}^2\right) \tag{18}$$

$$= 1 - \gamma \cdot \lim_{\rho_{z_p,y}\to 1} \left(\sum_{m'} \rho_{z_{m'},y}^2\right) \tag{19}$$

$$\lim_{\rho_{z_p,y}\to 1} \alpha = 1 - \gamma. \tag{20}$$

$\square$

# B  Limitations, Broader Impact and Future Work

While our proposed method offers several advantages for post-hoc ensemble selection, it is important to recognize its limitations. Unlike simpler ensembling heuristics, our approach requires tuning multiple training and architectural hyperparameters. Although we employed a fixed set of hyperparameters across all modalities and tasks in our experiments, this robustness may not generalize to all new tasks. In such cases, hyperparameter optimization may be necessary to achieve optimal performance. However, this could also enhance the results presented in this paper. Additionally, some bayesian approaches (McEwen et al., 2021) could further increase the robustness to the size of the validation data set.

Our approach is highly versatile and can be seamlessly integrated into a wide variety of ensemble-based learning systems, significantly enhancing their predictive capabilities. Because our method is agnostic to both the task and modality, we do not expect any inherent negative societal impacts. Instead, its effects will largely depend on how it is applied within different contexts and domains, making its societal implications contingent on the specific use case. In the future, we aim to explore in-context learning (Brown et al., 2020), where a pretrained Neural Ensembler could generate base model weights at test time, using their predictions as contextual input. We discuss broader impact and limitations in Appendix B.

# C  Space complexity of Neural Ensemblers

In this section, we discuss the memory advantage of the parameter-sharing mechanism by comparing it with a version without parameter-sharing. Take an instance $x$, such as an image or a text,

and consider a set of predictions from the $M$ base models for C classes $z = \{z_1^{(1)}, ..., z_M^{(1)}, ..., z_M^{(C)}\}$. If we vectorize these predictions, then $z \in \mathbb{R}^{C \cdot M}$. Using a two-layer neural network with hidden size $H$ and without parameter sharing (stacking mode) will demand $C \cdot M \cdot H$ parameters in the first layer and $H \cdot C$ parameters in the last layer. Hence, the space complexity is $\mathcal{O}(C \cdot M + M)$. On the other hand, our parameter-sharing neural ensembler consist in a neural network that receives as input $z^{(c)} = \{z_1^{(c)}, ..., z_M^{(c)}\}$, i.e. only the predictions for class $c$. This means that the first and second layer would have $M \cdot H$ and $H$ neurons respectively, with a space complexity $\mathcal{O}(M)$.

## D Related Work Addendum

**Ensemble Search via Bayesian Optimization**. Ensembles of models with different hyperparameters can be built using Bayesian optimization by iteratively swapping a model inside an ensemble with another one that maximizes the expected improvement (Levesque et al., 2016). DivBO (Shen et al., 2022) and subsequent work (Poduval et al., 2024) combine the ensemble's performance and diversity as a measure for expected improvement. Besides Bayesian Optimization, an evolutionary search can find robust ensembles of deep learning models (Zaidi et al., 2021). Although these approaches find optimal ensembles, they can overfit the validation data used for fitting if run for many iterations.

## E Baselines Details

We describe the setup for the baselines evaluation. Most of the baselines, except *random*, use the validation set for training the ensembling model or choosing the base models. For all the models that expected a fixed set of $N$ base models, we used $N = 50$, following previous work insights (Feurer et al., 2015). Some metadatasets (*Nasbench201-Mini,QuickTune-Mini* incurred in a large memory cost that made a few baselines fail in some datasets. Since this only happened in less than 10% of the datasets, we imputed the metrics for this specific cases, by using the metric of *single-best* as the imputation value. This mimics a real-world application scenario where the ensembler fall backs to just a *single-best* approach, which uses few memory for ensembling.

- **Single best** selects the best model according to the validation metric;

- **Random** chooses 50 random models to ensemble;

- **Top-N** ensembles the best $N \in \{5, 50\}$ models according to the validation metrics;

- **Greedy** (Caruana et al., 2004) creates an ensemble with $N = 50$ models by iterative selecting the one that improves the metric as proposed by previous work;

- **Quick** builds the ensemble with 50 models by adding model subsequently only if they strictly improve the metric;

- **CMAES** (Purucker and Beel, 2023b) uses an evolutive strategy with a post-processing method for ensembling. We used the implementation in the *Phem* library (Purucker, 2024);

- **Model Average** computed the weighted sum of the predictions as in Equation 3. We learn the weight using gradient-descent for optimizing the validation metric;

- **DivBO** (Shen et al., 2022): uses Bayesian Optimization for querying the best ensemble, by applying an acquisition function that accounts for both the diversity and performance. We implemented the method following the setup described by the authors in their paper;

- **Ensemble Optimization** (Levesque et al., 2016): uses Bayesian Optimization for selecting iteratively the model to replace another model inside the ensemble. We implemented the method following the default setup described by the authors;

Table 3: Average Ranked NLL

| | FTC | NB-Micro | NB-Mini | QT-Micro | QT-Mini | TR-Class | TR-Reg |
|---|---|---|---|---|---|---|---|
| **Single-Best** | $17.6667_{\pm0.9832}$ | $16.6667_{\pm2.3094}$ | $15.3333_{\pm2.7538}$ | $7.0167_{\pm2.5103}$ | $7.5333_{\pm2.5221}$ | $8.6325_{\pm4.6487}$ | $9.0294_{\pm4.5705}$ |
| **Random** | $20.0000_{\pm0.0000}$ | $9.3333_{\pm5.6862}$ | $12.3333_{\pm2.3094}$ | $19.0333_{\pm1.3060}$ | $19.0167_{\pm0.9143}$ | $14.0301_{\pm5.0166}$ | $16.4118_{\pm2.9803}$ |
| **Top5** | $13.6667_{\pm3.2660}$ | $11.6667_{\pm1.5275}$ | $8.3333_{\pm1.5275}$ | $6.9000_{\pm1.9360}$ | $8.5667_{\pm2.3146}$ | $7.0542_{\pm4.1083}$ | $8.4118_{\pm5.1455}$ |
| **Top50** | $13.3333_{\pm1.7512}$ | $5.3333_{\pm1.5275}$ | $5.0000_{\pm1.0000}$ | $13.9667_{\pm2.0212}$ | $13.9500_{\pm2.0776}$ | $7.6747_{\pm3.6729}$ | $8.1176_{\pm4.4984}$ |
| **Quick** | $7.6667_{\pm1.9664}$ | $7.3333_{\pm3.7859}$ | $\mathit{4.3333_{\pm3.5119}}$ | $4.7333_{\pm2.0331}$ | $6.1333_{\pm2.7131}$ | $6.9036_{\pm3.4273}$ | $\mathit{7.0000_{\pm4.1231}}$ |
| **Greedy** | $4.5000_{\pm1.3784}$ | $\mathit{4.3333_{\pm3.2146}}$ | $12.1667_{\pm2.7538}$ | $\mathit{3.5167_{\pm1.7786}}$ | $\mathit{4.7000_{\pm2.5278}}$ | $\mathit{6.6506_{\pm3.6138}}$ | $7.8824_{\pm4.1515}$ |
| **CMAES** | $15.3333_{\pm5.5737}$ | $16.6667_{\pm2.3094}$ | $15.3333_{\pm2.7538}$ | $15.8667_{\pm4.0809}$ | $17.4000_{\pm2.1066}$ | $11.4578_{\pm4.7094}$ | $7.3529_{\pm2.9356}$ |
| **Random Forest** | $8.6667_{\pm2.7325}$ | $15.6667_{\pm3.0551}$ | $17.0000_{\pm1.7321}$ | $15.0000_{\pm1.5702}$ | $11.1500_{\pm4.9674}$ | $13.3614_{\pm6.1159}$ | $9.6471_{\pm5.5895}$ |
| **Gradient Boosting** | $\mathit{4.0000_{\pm5.4037}}$ | $17.3333_{\pm3.0551}$ | $15.8333_{\pm3.6171}$ | $11.9667_{\pm6.8857}$ | $12.3000_{\pm6.8778}$ | $13.2771_{\pm5.8400}$ | $9.6471_{\pm5.4076}$ |
| **SVM** | $13.3333_{\pm1.6330}$ | $11.6667_{\pm8.5049}$ | $13.0000_{\pm8.6603}$ | $17.5333_{\pm1.5533}$ | $14.4500_{\pm6.5828}$ | $12.4639_{\pm5.0827}$ | $14.5882_{\pm6.9826}$ |
| **Linear** | $9.3333_{\pm2.1602}$ | $13.0000_{\pm2.6458}$ | $13.0000_{\pm3.6056}$ | $7.0000_{\pm3.1073}$ | $6.7833_{\pm3.4433}$ | $11.8434_{\pm6.0897}$ | $17.8824_{\pm4.4565}$ |
| **XGBoost** | $13.1667_{\pm5.4559}$ | $12.6667_{\pm4.1633}$ | $14.0000_{\pm6.2450}$ | $14.1333_{\pm2.1413}$ | $12.2500_{\pm3.0562}$ | $16.9819_{\pm3.4698}$ | $16.1765_{\pm3.7953}$ |
| **CatBoost** | $\mathbf{3.5000_{\pm3.3317}}$ | $15.0000_{\pm2.6458}$ | $15.6667_{\pm2.3094}$ | $11.7000_{\pm1.8919}$ | $12.9667_{\pm3.5548}$ | $11.3012_{\pm4.8583}$ | $10.4118_{\pm4.8484}$ |
| **LightGBM** | $10.6667_{\pm6.1860}$ | $17.0000_{\pm5.1962}$ | $17.0000_{\pm5.1962}$ | $13.4333_{\pm2.8093}$ | $15.1333_{\pm2.8945}$ | $17.6867_{\pm3.7900}$ | $13.2353_{\pm5.2978}$ |
| **Akaike** | $13.5000_{\pm3.3166}$ | $5.0000_{\pm1.0000}$ | $4.6667_{\pm0.5774}$ | $12.5333_{\pm2.0083}$ | $12.6833_{\pm1.9761}$ | $7.9639_{\pm4.0166}$ | $7.3824_{\pm4.6788}$ |
| **MA** | $14.1667_{\pm3.8687}$ | $8.0000_{\pm1.7321}$ | $5.0000_{\pm3.4641}$ | $16.7667_{\pm1.4003}$ | $16.1000_{\pm2.1270}$ | $10.3916_{\pm4.9226}$ | $9.9412_{\pm6.5141}$ |
| **DivBO** | $8.5000_{\pm4.5497}$ | $11.0000_{\pm6.2450}$ | $\mathit{4.3333_{\pm4.0415}}$ | $4.9500_{\pm2.5876}$ | $5.2000_{\pm2.7687}$ | $7.4337_{\pm3.5585}$ | $7.8824_{\pm3.6552}$ |
| **EO** | $7.1667_{\pm4.5789}$ | $6.0000_{\pm2.0000}$ | $9.0000_{\pm2.6458}$ | $5.8833_{\pm2.2194}$ | $5.6167_{\pm2.5484}$ | $7.0241_{\pm3.4144}$ | $8.2353_{\pm3.4192}$ |
| **NE-Stack** | $7.6667_{\pm5.9889}$ | $\mathbf{1.0000_{\pm0.0000}}$ | $4.0000_{\pm5.1962}$ | $3.3000_{\pm3.6874}$ | $2.4000_{\pm2.3282}$ | $11.7470_{\pm7.1326}$ | $15.2353_{\pm4.6169}$ |
| **NE-MA** | $4.1667_{\pm3.3116}$ | $5.3333_{\pm3.5119}$ | $4.6667_{\pm2.3094}$ | $4.7667_{\pm2.1445}$ | $5.6667_{\pm2.1389}$ | $6.1205_{\pm3.8553}$ | $5.5294_{\pm2.7184}$ |

- **Machine Learning Models as Stackers**: We used the default configurations provided by Sckit-learn (Pedregosa et al., 2011) for these stackers. The input to the models is the concatenation of all the base models' predictions. We concatenated the probabilistic predictions from all the classes in the classification tasks. This sometimes produced a large dimensional input space, and a large memory load. The model was trained on the validation set. Specifically, we apply popular models: *SVM*, *Random Forest*, *Gradient Boosting*, *Linear Regression*, *XGBoost*, *LightGBM*;

- **Akaike Weighting or Pseudo Model Averaging** (Wagenmakers and Farrell, 2004) computes weights using the relative performance of every model. Assuming that the lowest metric from the pool of base models is $\ell_{\min} \in \{\ell_1, ..., \ell_M\}$, then the weight is computed using $\Delta_i = \ell_i - \ell_{\min}$ as:

$$w_i = \frac{\exp^{-\frac{1}{2}\Delta_i}}{\sum_m \exp^{-\frac{1}{2}\Delta_m}} \tag{21}$$

- **Dynamic Ensemble Search Baselines** learn a model that ensembles different models per instance. We used baselines from the *DESlib* library (Cruz et al., 2020) such as **KNOP** (Cavalin et al., 2013), **KNORAE** (Ko et al., 2008) and **MetaDES** (Cruz et al., 2015).

## F Additional Results Related to Research Question #1

In this Section we report additional results from our experiments:

- Average Ranking of baselines for the Negative Log-likelihood (Table 3) and Classification Errors (Table 4).

- Average NLL in *Scikit-learn Pipelines* metadataset (Figure 6).

Table 4: Average Ranked Error

| | FTC | NB-Micro | NB-Mini | QT-Micro | QT-Mini | TR-Class | TR-Class (AUC) |
|---|---|---|---|---|---|---|---|
| **Single-Best** | $14.6667_{\pm3.4593}$ | $18.3333_{\pm0.2887}$ | $17.1667_{\pm1.5275}$ | $11.0167_{\pm4.8128}$ | $9.7333_{\pm4.0231}$ | $10.7048_{\pm5.2452}$ | $11.2840_{\pm4.3236}$ |
| **Random** | $19.3333_{\pm1.6330}$ | $14.6667_{\pm1.1547}$ | $14.8333_{\pm0.7638}$ | $19.8667_{\pm0.3198}$ | $19.6667_{\pm0.6065}$ | $13.8193_{\pm6.5698}$ | $15.0494_{\pm5.8130}$ |
| **Top5** | $6.5833_{\pm4.4768}$ | $13.3333_{\pm2.0817}$ | $10.6667_{\pm3.5119}$ | $4.8833_{\pm3.3725}$ | $4.8667_{\pm3.3859}$ | $10.5181_{\pm4.6751}$ | $8.5741_{\pm5.0013}$ |
| **Top50** | $16.5000_{\pm3.3317}$ | $4.3333_{\pm1.5275}$ | $2.3333_{\pm1.5275}$ | $9.6333_{\pm4.9444}$ | $10.9000_{\pm3.2094}$ | $9.6988_{\pm5.1234}$ | $8.6605_{\pm4.7189}$ |
| **Quick** | $6.8333_{\pm4.8442}$ | $8.6667_{\pm1.1547}$ | $7.3333_{\pm2.3094}$ | $6.1000_{\pm4.0480}$ | $3.6333_{\pm2.4066}$ | $10.0301_{\pm4.0412}$ | $7.9136_{\pm4.9319}$ |
| **Greedy** | $5.5833_{\pm4.3865}$ | $13.0000_{\pm3.4641}$ | $11.3333_{\pm1.5275}$ | $10.1167_{\pm4.3543}$ | $7.5000_{\pm3.7093}$ | $9.7108_{\pm4.1069}$ | $8.1975_{\pm4.5830}$ |
| **CMAES** | $5.0833_{\pm2.3752}$ | $6.0000_{\pm3.6056}$ | $7.3333_{\pm2.0817}$ | $9.6333_{\pm3.9105}$ | $6.6167_{\pm2.6152}$ | $10.5060_{\pm4.8341}$ | $11.6173_{\pm6.0220}$ |
| **Random Forest** | $7.1667_{\pm4.6224}$ | $9.0000_{\pm5.1962}$ | $11.1667_{\pm5.5752}$ | $13.8000_{\pm4.5837}$ | $14.3667_{\pm2.7697}$ | $10.5542_{\pm4.6302}$ | $10.6667_{\pm5.9119}$ |
| **Gradient Boosting** | $12.5000_{\pm4.4159}$ | $19.0000_{\pm0.8660}$ | $17.6667_{\pm2.2546}$ | $9.6667_{\pm5.1551}$ | $14.2333_{\pm4.1351}$ | $11.0181_{\pm5.9671}$ | $11.0185_{\pm6.6635}$ |
| **SVM** | $3.8333_{\pm3.3714}$ | $9.0000_{\pm1.7321}$ | $10.5000_{\pm5.8949}$ | $12.0000_{\pm6.0955}$ | $15.6500_{\pm3.5261}$ | $8.3855_{\pm4.7788}$ | $13.0185_{\pm7.1144}$ |
| **Linear** | $8.8333_{\pm3.1252}$ | $13.6667_{\pm5.0332}$ | $14.1667_{\pm3.6856}$ | $6.2500_{\pm3.5834}$ | $7.9333_{\pm3.2872}$ | $9.3916_{\pm6.3745}$ | $10.9259_{\pm6.7519}$ |
| **XGBoost** | $11.4167_{\pm5.5355}$ | $14.6667_{\pm2.3094}$ | $13.0000_{\pm1.7321}$ | $17.0000_{\pm3.2350}$ | $16.1333_{\pm3.0932}$ | $12.5120_{\pm5.8304}$ | $11.6111_{\pm4.4651}$ |
| **CatBoost** | $14.6667_{\pm3.3267}$ | $13.0000_{\pm0.0000}$ | $17.5000_{\pm1.5000}$ | $12.0500_{\pm3.8155}$ | $14.5167_{\pm3.2310}$ | $11.9699_{\pm5.5179}$ | $11.4506_{\pm4.2834}$ |
| **LightGBM** | $10.5833_{\pm5.0241}$ | $17.0000_{\pm5.1962}$ | $17.0000_{\pm5.1962}$ | $15.7833_{\pm3.4433}$ | $17.5167_{\pm2.4193}$ | $11.7470_{\pm6.1947}$ | $11.4815_{\pm4.6094}$ |
| **Akaike** | $13.3333_{\pm6.6608}$ | $2.6667_{\pm2.0817}$ | $3.3333_{\pm0.5774}$ | $9.1500_{\pm4.8551}$ | $10.1500_{\pm3.5016}$ | $9.9639_{\pm5.0677}$ | $11.7654_{\pm4.2925}$ |
| **MA** | $16.0000_{\pm3.4641}$ | $5.3333_{\pm1.1547}$ | $2.6667_{\pm2.0817}$ | $11.6833_{\pm4.3519}$ | $12.7833_{\pm3.7132}$ | $10.1024_{\pm5.3487}$ | $10.4815_{\pm4.7994}$ |
| **DivBO** | $11.2500_{\pm4.8760}$ | $12.3333_{\pm1.1547}$ | $15.6667_{\pm2.7538}$ | $11.2167_{\pm3.8117}$ | $7.9000_{\pm3.8336}$ | $9.8193_{\pm4.5830}$ | $10.1667_{\pm4.6657}$ |
| **EO** | $12.6667_{\pm5.7504}$ | $5.3333_{\pm3.7859}$ | $4.6667_{\pm2.5166}$ | $10.5167_{\pm3.4851}$ | $9.0667_{\pm3.0618}$ | $9.9036_{\pm5.3905}$ | $8.8889_{\pm4.0226}$ |
| **NE-Stack** | $5.5000_{\pm3.3317}$ | $4.0000_{\pm1.7321}$ | $8.3333_{\pm0.5774}$ | $3.7500_{\pm2.7189}$ | $2.7833_{\pm2.2995}$ | $10.8313_{\pm6.6403}$ | $9.3148_{\pm7.0986}$ |
| **NE-MA** | $7.6667_{\pm2.2509}$ | $2.6667_{\pm1.5275}$ | $3.3333_{\pm2.5166}$ | $5.8833_{\pm4.3225}$ | $4.0500_{\pm2.4046}$ | $8.8133_{\pm4.9137}$ | $7.9136_{\pm4.8456}$ |

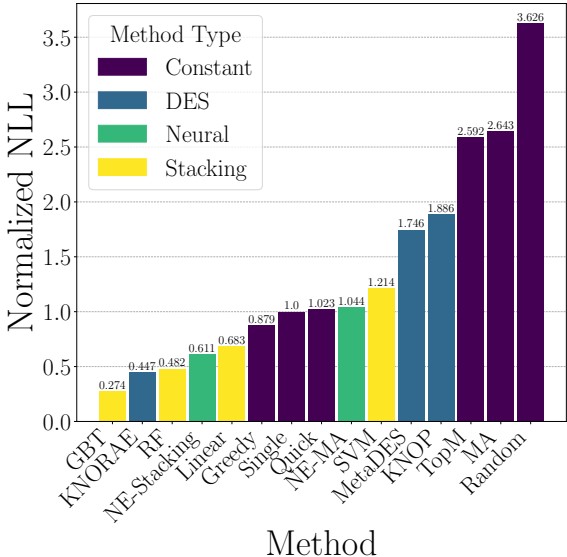

Figure 6: Average Normalized NLL in Scikit-learn pipelines (Validation).

## F.1 Unnormalized Results

We report the results for the research question 1 in Tables 5 and 6, omitting the normalization. This helps us to understand how much the normalization is changing the results. However, as the different datasets have metrics with different scales, it is important to normalize the results to get a better picture of the relative performances. This is especially problematic in the regression tasks, as it depends on the target scale, therefore we omit it. Even without the normalization, our proposed approach achieves the best results across different datasets. The ranking results remain the same independently of the normalization.

Table 5: Average Unnormalized NLL.

| | FTC | NB-Micro | NB-Mini | QT-Micro | QT-Mini | TR-Class |
|---|---|---|---|---|---|---|
| **Single-Best** | $0.3412_{\pm 0.3043}$ | $1.9175_{\pm 1.2203}$ | $1.5696_{\pm 0.8119}$ | $0.4499_{\pm 0.5107}$ | $0.6339_{\pm 0.5276}$ | $0.3647_{\pm 0.3300}$ |
| **Random** | $0.4521_{\pm 0.2982}$ | $1.2202_{\pm 1.0069}$ | $1.3005_{\pm 0.9988}$ | $2.1473_{\pm 0.6480}$ | $2.7139_{\pm 1.0686}$ | $0.4344_{\pm 0.3341}$ |
| **Top5** | $0.2975_{\pm 0.2799}$ | $1.2940_{\pm 0.9934}$ | $1.1911_{\pm 0.9833}$ | $0.4234_{\pm 0.4473}$ | $0.6702_{\pm 0.5330}$ | $0.3576_{\pm 0.3306}$ |
| **Top50** | $0.2819_{\pm 0.2631}$ | $1.1515_{\pm 0.9713}$ | _$1.1496_{\pm 0.9797}$_ | $0.7813_{\pm 0.5492}$ | $1.2587_{\pm 0.8605}$ | _$0.3561_{\pm 0.3331}$_ |
| **Quick** | $0.2443_{\pm 0.2128}$ | $1.1679_{\pm 0.9614}$ | **$1.1455_{\pm 0.9608}$** | $0.3952_{\pm 0.4441}$ | $0.6030_{\pm 0.4869}$ | $0.3562_{\pm 0.3326}$ |
| **Greedy** | $0.2366_{\pm 0.2107}$ | $1.1498_{\pm 0.9652}$ | $1.2953_{\pm 0.9657}$ | _$0.3895_{\pm 0.4453}$_ | _$0.5817_{\pm 0.4870}$_ | $0.3566_{\pm 0.3317}$ |
| **CMAES** | $0.3552_{\pm 0.2130}$ | $1.9175_{\pm 1.2203}$ | $1.5696_{\pm 0.8119}$ | $1.1665_{\pm 0.7614}$ | $1.8487_{\pm 0.6070}$ | $0.3862_{\pm 0.3325}$ |
| **Random Forest** | $0.2434_{\pm 0.2033}$ | $1.7354_{\pm 1.4580}$ | $1.6589_{\pm 1.4265}$ | $0.8846_{\pm 0.6550}$ | $0.9870_{\pm 0.8090}$ | $0.4452_{\pm 0.3734}$ |
| **Gradient Boosting** | **$0.2262_{\pm 0.1915}$** | $2.2979_{\pm 0.5750}$ | $1.8206_{\pm 0.6327}$ | $1.2480_{\pm 1.6045}$ | $1.7043_{\pm 1.9723}$ | $0.4708_{\pm 0.4264}$ |
| **SVM** | $0.2626_{\pm 0.2241}$ | $1.6698_{\pm 1.4024}$ | $1.6590_{\pm 1.4274}$ | $1.3608_{\pm 0.3953}$ | $1.8931_{\pm 1.2176}$ | $0.3951_{\pm 0.3356}$ |
| **Linear** | $0.2496_{\pm 0.2130}$ | $1.3021_{\pm 0.9815}$ | $1.3626_{\pm 0.9836}$ | $0.4701_{\pm 0.5151}$ | $0.6582_{\pm 0.6088}$ | $0.4095_{\pm 0.3914}$ |
| **XGBoost** | $0.2678_{\pm 0.2250}$ | $1.5484_{\pm 1.2487}$ | $1.5788_{\pm 1.2796}$ | $0.8328_{\pm 0.4807}$ | $0.9423_{\pm 0.5402}$ | $0.5503_{\pm 0.4738}$ |
| **CatBoost** | _$0.2329_{\pm 0.2074}$_ | $1.6813_{\pm 1.3589}$ | $1.6531_{\pm 1.3327}$ | $0.6392_{\pm 0.6006}$ | $1.0816_{\pm 0.7100}$ | $0.3954_{\pm 0.3839}$ |
| **LightGBM** | $0.2442_{\pm 0.2000}$ | $6.7760_{\pm 5.5841}$ | $6.7783_{\pm 5.5802}$ | $0.7957_{\pm 0.7383}$ | $2.5903_{\pm 3.4557}$ | $0.6301_{\pm 0.5723}$ |
| **Akaike** | $0.3012_{\pm 0.3010}$ | $1.1503_{\pm 0.9699}$ | $1.1499_{\pm 0.9822}$ | $0.7369_{\pm 0.5221}$ | $1.0655_{\pm 0.6000}$ | _$0.3561_{\pm 0.3334}$_ |
| **MA** | $0.2804_{\pm 0.2074}$ | $1.1612_{\pm 0.9664}$ | $1.1530_{\pm 0.9720}$ | $1.0974_{\pm 0.5102}$ | $1.5240_{\pm 0.8090}$ | $0.3788_{\pm 0.3342}$ |
| **DivBO** | $0.2643_{\pm 0.2526}$ | $1.2355_{\pm 0.8746}$ | $1.1659_{\pm 0.9729}$ | $0.4065_{\pm 0.4489}$ | $0.5906_{\pm 0.4825}$ | $0.3588_{\pm 0.3339}$ |
| **EO** | $0.2599_{\pm 0.2481}$ | $1.1532_{\pm 0.9801}$ | $1.2024_{\pm 0.9656}$ | $0.4144_{\pm 0.4542}$ | $0.5937_{\pm 0.4804}$ | $0.3572_{\pm 0.3329}$ |
| **NE-Stack** | $0.2375_{\pm 0.2031}$ | **$1.0706_{\pm 0.9554}$** | $1.1543_{\pm 1.0696}$ | **$0.3747_{\pm 0.4494}$** | **$0.4923_{\pm 0.4658}$** | $0.4587_{\pm 0.4479}$ |
| **NE-MA** | $0.2399_{\pm 0.2176}$ | _$1.1486_{\pm 0.9892}$_ | $1.1596_{\pm 0.9991}$ | $0.3998_{\pm 0.4474}$ | $0.5832_{\pm 0.4900}$ | **$0.3536_{\pm 0.3311}$** |

Table 6: Average Unnormalized Error.

| | FTC | NB-Micro | NB-Mini | QT-Micro | QT-Mini | TR-Class | TR-Class (AUC) |
|---|---|---|---|---|---|---|---|
| **Single-Best** | $0.0868_{\pm 0.0853}$ | $0.4320_{\pm 0.3331}$ | $0.4509_{\pm 0.3019}$ | $0.1285_{\pm 0.1499}$ | $0.1667_{\pm 0.1477}$ | $0.1604_{\pm 0.1545}$ | $0.1169_{\pm 0.1164}$ |
| **Random** | $0.1012_{\pm 0.0910}$ | $0.3092_{\pm 0.2444}$ | $0.3502_{\pm 0.2417}$ | $0.5677_{\pm 0.2487}$ | $0.5295_{\pm 0.2306}$ | $0.1840_{\pm 0.1632}$ | $0.1422_{\pm 0.1356}$ |
| **Top5** | $0.0831_{\pm 0.0825}$ | $0.3005_{\pm 0.2374}$ | $0.3007_{\pm 0.2251}$ | _$0.1079_{\pm 0.1375}$_ | $0.1502_{\pm 0.1359}$ | $0.1594_{\pm 0.1527}$ | **$0.1125_{\pm 0.1142}$** |
| **Top50** | $0.0881_{\pm 0.0829}$ | $0.2778_{\pm 0.2266}$ | $0.2788_{\pm 0.2262}$ | $0.1334_{\pm 0.1495}$ | $0.1816_{\pm 0.1555}$ | $0.1584_{\pm 0.1524}$ | _$0.1139_{\pm 0.1160}$_ |
| **Quick** | $0.0831_{\pm 0.0824}$ | $0.2840_{\pm 0.2304}$ | $0.2842_{\pm 0.2289}$ | $0.1121_{\pm 0.1409}$ | $0.1504_{\pm 0.1425}$ | $0.1583_{\pm 0.1517}$ | $0.1145_{\pm 0.1192}$ |
| **Greedy** | $0.0824_{\pm 0.0811}$ | $0.3253_{\pm 0.2827}$ | $0.2958_{\pm 0.2368}$ | $0.1251_{\pm 0.1487}$ | $0.1596_{\pm 0.1450}$ | $0.1562_{\pm 0.1523}$ | $0.1140_{\pm 0.1186}$ |
| **CMAES** | _$0.0822_{\pm 0.0810}$_ | $0.2788_{\pm 0.2247}$ | $0.2869_{\pm 0.2358}$ | $0.1234_{\pm 0.1478}$ | $0.1579_{\pm 0.1456}$ | $0.1589_{\pm 0.1541}$ | $0.1269_{\pm 0.1311}$ |
| **Random Forest** | $0.0833_{\pm 0.0829}$ | $0.2958_{\pm 0.2444}$ | $0.3706_{\pm 0.3698}$ | $0.1666_{\pm 0.1955}$ | $0.2122_{\pm 0.1874}$ | $0.1591_{\pm 0.1528}$ | $0.1198_{\pm 0.1210}$ |
| **Gradient Boosting** | $0.0838_{\pm 0.0809}$ | $0.4682_{\pm 0.2852}$ | $0.4920_{\pm 0.2559}$ | $0.1425_{\pm 0.1773}$ | $0.2092_{\pm 0.1855}$ | $0.1620_{\pm 0.1592}$ | $0.1240_{\pm 0.1293}$ |
| **SVM** | **$0.0819_{\pm 0.0806}$** | $0.2917_{\pm 0.2431}$ | $0.3356_{\pm 0.2444}$ | $0.1627_{\pm 0.2031}$ | $0.2341_{\pm 0.2032}$ | _$0.1553_{\pm 0.1550}$_ | $0.1407_{\pm 0.1358}$ |
| **Linear** | $0.0830_{\pm 0.0816}$ | $0.3098_{\pm 0.2200}$ | $0.3469_{\pm 0.2097}$ | $0.1191_{\pm 0.1503}$ | $0.1670_{\pm 0.1547}$ | $0.1577_{\pm 0.1568}$ | $0.1195_{\pm 0.1203}$ |
| **XGBoost** | $0.0848_{\pm 0.0840}$ | $0.3546_{\pm 0.2824}$ | $0.3554_{\pm 0.2836}$ | $0.1903_{\pm 0.1452}$ | $0.2144_{\pm 0.1598}$ | $0.1654_{\pm 0.1574}$ | $0.1193_{\pm 0.1199}$ |
| **CatBoost** | $0.0874_{\pm 0.0863}$ | $0.4022_{\pm 0.3034}$ | $0.4217_{\pm 0.3411}$ | $0.1450_{\pm 0.1754}$ | $0.2090_{\pm 0.1822}$ | $0.1629_{\pm 0.1606}$ | $0.1199_{\pm 0.1224}$ |
| **LightGBM** | $0.0843_{\pm 0.0833}$ | $0.5941_{\pm 0.4476}$ | $0.5907_{\pm 0.4475}$ | $0.1666_{\pm 0.1604}$ | $0.3442_{\pm 0.3146}$ | $0.1633_{\pm 0.1586}$ | $0.1196_{\pm 0.1222}$ |
| **Akaike** | $0.0857_{\pm 0.0802}$ | **$0.2767_{\pm 0.2252}$** | $0.2797_{\pm 0.2273}$ | $0.1311_{\pm 0.1487}$ | $0.1769_{\pm 0.1538}$ | $0.1589_{\pm 0.1527}$ | $0.1228_{\pm 0.1305}$ |
| **MA** | $0.0874_{\pm 0.0803}$ | $0.2795_{\pm 0.2271}$ | **$0.2765_{\pm 0.2248}$** | $0.1435_{\pm 0.1618}$ | $0.1971_{\pm 0.1651}$ | $0.1609_{\pm 0.1580}$ | $0.1186_{\pm 0.1262}$ |
| **DivBO** | $0.0846_{\pm 0.0832}$ | $0.3024_{\pm 0.2489}$ | $0.3817_{\pm 0.1893}$ | $0.1295_{\pm 0.1431}$ | $0.1616_{\pm 0.1458}$ | $0.1573_{\pm 0.1540}$ | $0.1170_{\pm 0.1200}$ |
| **EO** | $0.0851_{\pm 0.0829}$ | $0.2786_{\pm 0.2283}$ | $0.2813_{\pm 0.2274}$ | $0.1330_{\pm 0.1561}$ | $0.1659_{\pm 0.1471}$ | $0.1589_{\pm 0.1577}$ | $0.1177_{\pm 0.1237}$ |
| **NE-Stack** | $0.0824_{\pm 0.0817}$ | $0.2781_{\pm 0.2275}$ | $0.2879_{\pm 0.2339}$ | **$0.1033_{\pm 0.1348}$** | **$0.1424_{\pm 0.1314}$** | $0.1607_{\pm 0.1511}$ | $0.1152_{\pm 0.1159}$ |
| **NE-MA** | $0.0828_{\pm 0.0818}$ | _$0.2772_{\pm 0.2264}$_ | _$0.2773_{\pm 0.2262}$_ | $0.1093_{\pm 0.1344}$ | _$0.1490_{\pm 0.1383}$_ | **$0.1551_{\pm 0.1515}$** | $0.1149_{\pm 0.1211}$ |

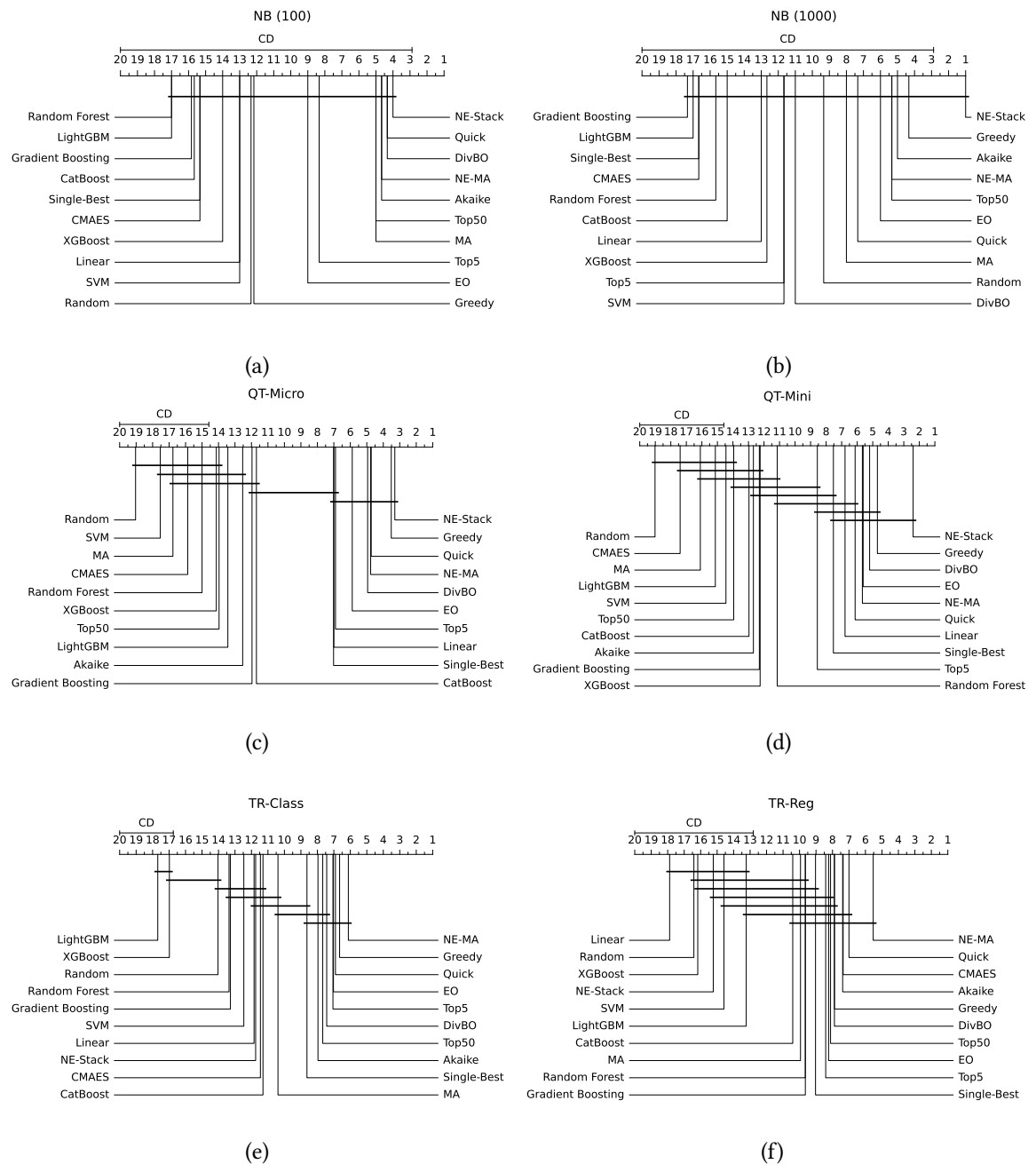

Figure 7: Critical Difference (CD) diagrams aggregating datasets across metadatasets

# G  Critical Difference Diagrams

We want to gain deeper insights into the difference between our proposed method and the baselines.
**Critical Difference Diagrams**. To this end, we present critical difference (CD) diagrams to visualize and statistically analyze the performance of different methods across multiple datasets. The CD diagrams are generated using the `autorank` library[3], which automates the statistical comparison by employing non-parametric tests like Friedman test followed by the Nemenyi post-hoc test. These

---

[3]`https://github.com/sherbold/autorank`

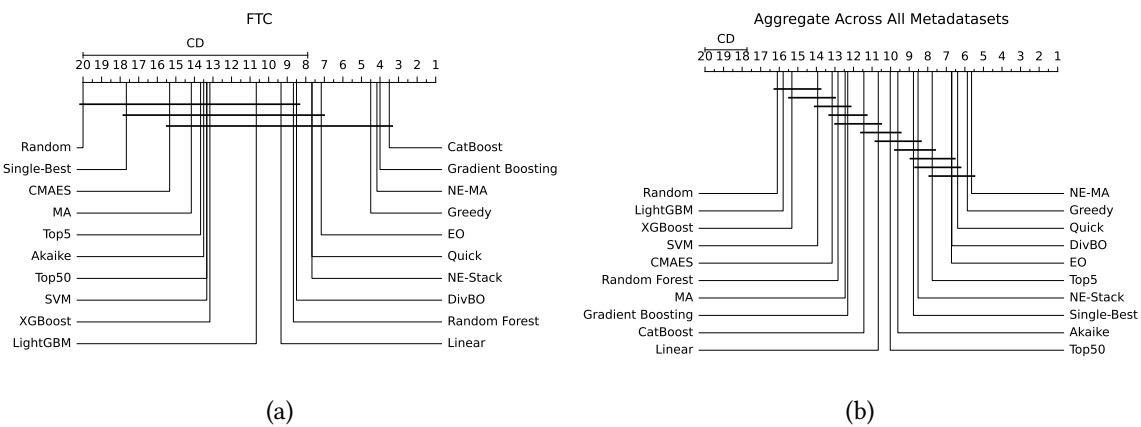

Figure 8: Critical Difference (CD) diagrams aggregating datasets across metadatasets

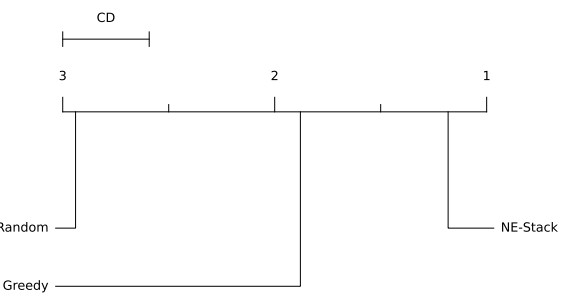

Figure 9: CD Diagram in datasets with a lot of classes.

diagrams show the average ranks of the methods along the horizontal axis, where methods are positioned based on their performance across datasets. The critical difference value, which depends on the number of datasets, is represented by a horizontal bar above the ranks. Methods not connected by this bar exhibit statistically significant differences in performance.

**Evaluation**. We computed CD diagrams across all datasets in all metadatasets (Figures 7a-8a) and an aggregated diagram across all datasets in all metadatasets (Figure 8b). We observe that although the performance difference is not substantial compared to other top-performing post-hoc ensembles like *Greedy* and *Quick*, our Neural Ensembler (**NE-MA**) consistently achieves the best performance in the aggregated results (Figure 8b). We also highlight that the **NE** versions were the top-performing approaches across all metadatasets except FTC, even though we did not modify the method's hyperparameters.

**Significance on Challenging Datasets**. Given the high performance between *Greedy* and **NE-MA**, we wanted to understand when the second one would obtain strong significant results. We found that that **NE-MA** is particularly well performing in challenging datasets with a large number of classes. Given Table 11, we can see that four meta-datasets have a high (> 10) number of classes, thus they have datasets with a lot of classes. We selected these metadatasets (*NB(100), NB(1000), QT-Micro, QT-Mini*), and plotted the significance compared to *Greedy* and *Random Search*. The results reported in Figure 9 demonstrate that our approach is significantly better than Greedy in these metadatasets.

## H  Empirical Analysis of Computational Cost

To better understand the trade-off between performance and computational cost, we analyze the average normalized Negative Log-Likelihood (NLL) and the runtime of various ensembling techniques. Our focus is on post-hoc ensembling methods, assuming that all base models are pre-trained. This allows us to isolate and compare the efficiency of the ensembling processes themselves, in contrast to methods like DivBO, which sequentially train models during the search, leading to higher computational demands. We measure only the post-hoc ensembling time, i.e. training the ensembler or selecting the group of base models for the final ensemble.

As shown in Figure 10, the Neural Ensemblers (**NE-MA** and **NE-Stack**) achieve the best average performance while maintaining a competitive runtime. Notably, our method has a shorter runtime than the *Greedy* ensembling method and surpasses it in terms of performance. Additionally, the Neural Ensemblers are faster than traditional machine learning models such as *Gradient Boosting*, *Support Vector Machines (SVM)*, *Random Forests*, and optimization algorithms like *CMAES*. While simpler methods like *Top5* and *Top50* exhibit faster runtimes, they do so at the expense of reduced accuracy.

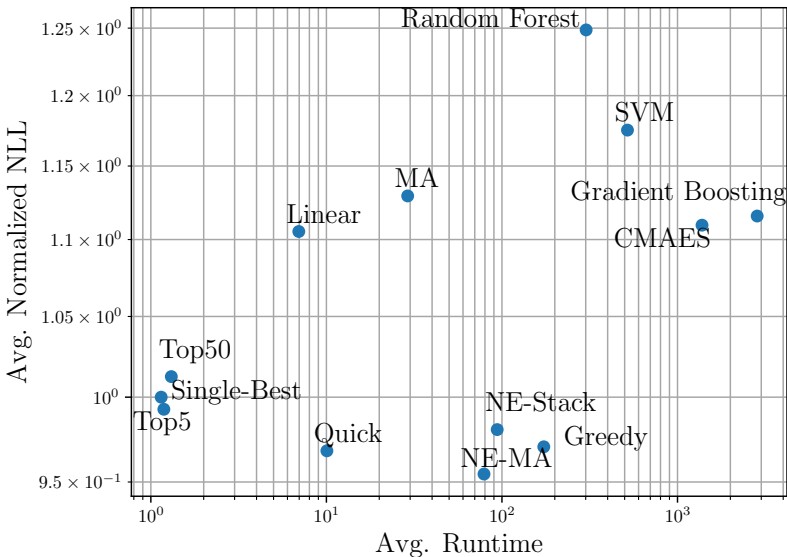

Figure 10: Trade-off between performance and computational cost for different ensembling methods. The Neural Ensemblers achieve the best performance with competitive runtime, outperforming other methods in both accuracy and efficiency.

## I  Proof-of-Concept with Overparameterized Base Models

A key question is whether dynamic ensemblers like our Neural Ensembler (NE) offer benefits when base models are overparameterized and potentially overfit the data. Specifically, does the advantage of the NE persist in scenarios where the base models have high capacity?

To explore this, we extend our Proof-of-Concept experiment by ensembling 10th-degree polynomials instead of 2nd-degree ones, thereby increasing the complexity of the base models and introducing the risk of overfitting. We follow the same protocol as in Section 2.1, comparing the performance of the NE with the fixed-weight ensemble method.

As illustrated in Figure 11, even with overparameterized base models that tend to overfit the training data, the NE (specifically the weighted Model-Averaging version) achieves the best performance on unseen data. This improvement occurs because the NE dynamically adjusts the ensemble weights based on the input. Thus, dynamic ensembling is advantageous not only when

base models underfit but also when they overfit the data. In contrast, the fixed-weight approach lacks this adaptability and cannot compensate for the overfitting behavior of the base models.

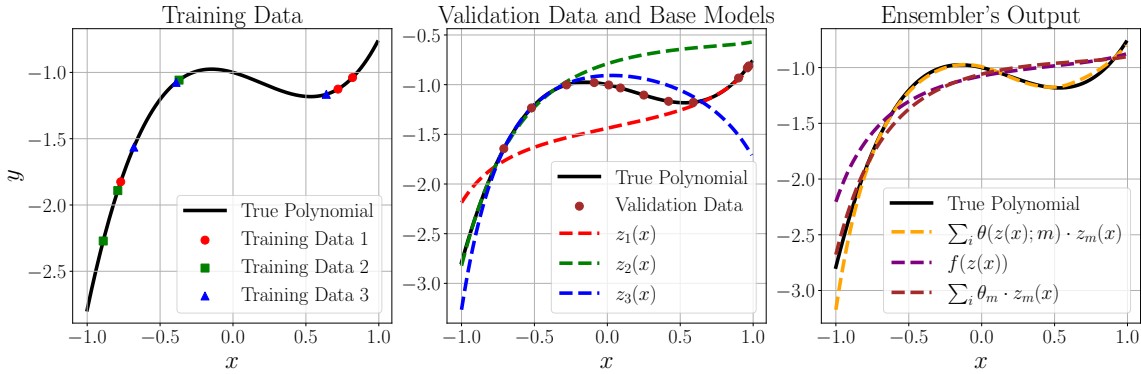

Figure 11: Proof-of-Concept experiment with overparameterized base models (10th-degree polynomials). The Neural Ensembler outperforms fixed-weight ensemble method by mitigating overfitting through dynamic, input-dependent weighting.

## J Ablation Study on Validation Data Size

To assess the Neural Ensembler's (NE) dependence on validation data size and its sample efficiency, we conducted an ablation study by varying the proportion of validation data used for training.

In this study, we evaluate the NE in stacking mode across all metadatasets, using different percentages of the available validation data: 1%, 5%, 10%, 25%, 50%, 100%. For each configuration, we compute the Negative Log Likelihood (NLL) on the test set and normalize these values by the performance achieved when using 100% of the validation data. This normalization allows us to compare performance changes across different tasks, accounting for differences in metric scales. A normalized NLL value below 1 indicates performance better than the baseline with full validation data, while a value above 1 indicates a performance drop.

The results are presented in Figure 12. We observe that reducing the amount of validation data used for training the NE leads to a relative degradation of the performance. However, the performance drops are relatively modest in three metadatasets. For these experiments we used the same dropout rate 0.75. We could improve the robustness in lower percentages of validation data by using a higher dropout rate. The DropOut mechanism prevents overfitting by randomly omitting base models during training, while parameter sharing reduces the number of parameters and promotes learning common representations.

## K Effect of Merging Training and Validation Data

In our experimental setup (Section 4), we train the base models using the training split and the ensemblers using the validation split, then evaluate on the test split. An important question is whether merging the training and validation data could improve the performance of both the base models and the ensemblers. Specifically, we explore:

(a) Can baseline methods that do not require a validation split, such as *Random*, achieve better performance if the base models are trained on the merged dataset (training + validation)?

(b) Would training both the base models and the ensemblers on the merged dataset be beneficial, given that more data might enhance their learning?

To investigate these questions, we conducted experiments on two metadatasets: *Scikit-learn Pipelines* and *FTC*. We trained the base models on the merged dataset and also trained the ensemblers

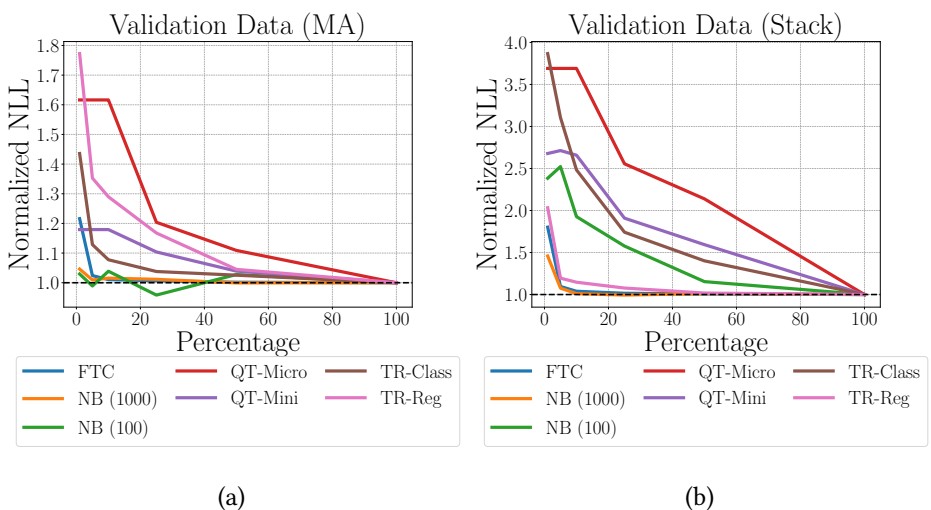

Figure 12: Ablation study on the percentage of validation data used for training the Neural Ensembler. The normalized NLL is plotted against the percentage of validation data, with values below 1 indicating performance better than or equal to using the full validation set.

on this same data. Five representative baselines were compared: **NE-MA**, **NE-Stack**, *Greedy*, *Random* and *Single-best*.

Figure 13b presents the test set performance on the *Scikit-learn Pipelines* metadataset. Training on the merged dataset did not improve performance compared to training on the original splits. In fact, the results are similar to the *Random* ensembling method, indicating no significant gain. This suggests that training both the base models and ensemblers on the same (larger) dataset may lead to **overfitting**, hindering generalization to unseen data.

Further evidence of overfitting is shown in Figure 13a, where we report the Negative Log Likelihood (NLL) on the merged dataset (effectively the training data). The NLL values are significantly lower for models trained on the merged dataset, confirming that they fit the training data well but do not generalize to the test set.

Similar observations were made on the *FTC* metadataset, as presented in Table 14a and Figure 14b. Although the *Random* method trained on the merged dataset shows a slight improvement, the overall performance gains are minimal.

From these results, we conclude that:

(a) Training base models on the merged dataset does not significantly enhance the performance of baseline methods that do not require a validation split.

(b) Using the merged dataset to train both the base models and the ensemblers is not beneficial and may lead to overfitting, reducing generalization to the test set.

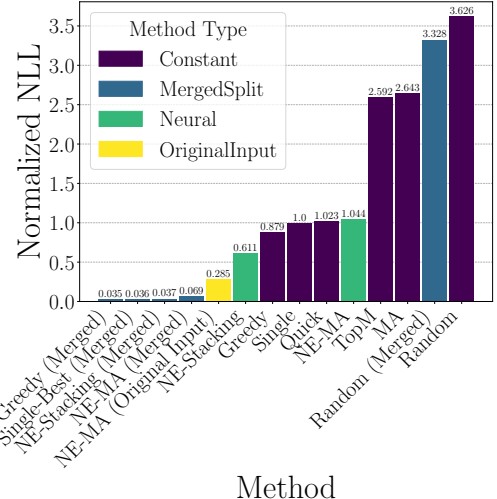
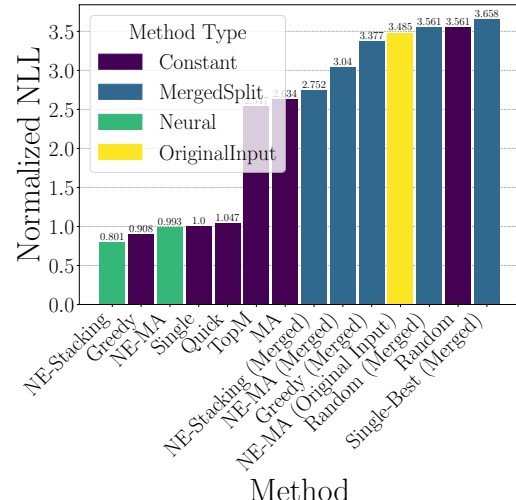

(a) Performance on merged dataset (training data) for *Scikit-learn Pipelines*.

(b) Test set performance on *Scikit-learn Pipelines* with merged training.

Figure 13: Impact of training on merged dataset for *Scikit-learn Pipelines*. Training on the merged dataset leads to overfitting, as evidenced by low NLL on training data but no improvement on test data.

| Algorithm | FTC (Avg. Normalized NLL) | FTC (Avg. Rank) |
|---|---|---|
| Single-Best | $1.0000_{\pm0.0000}$ | $11.4167_{\pm1.5626}$ |
| Single-Best (Merged) | $1.2816_{\pm0.6710}$ | $11.2500_{\pm4.0466}$ |
| Random | $1.5450_{\pm0.5289}$ | $14.0000_{\pm0.8944}$ |
| Random (Merged) | $1.5365_{\pm0.7142}$ | $13.0833_{\pm2.2454}$ |
| Top5 | $0.8406_{\pm0.0723}$ | $8.6667_{\pm2.9439}$ |
| Top50 | $0.8250_{\pm0.1139}$ | $8.3333_{\pm1.9664}$ |
| Quick | $0.7273_{\pm0.0765}$ | $4.6667_{\pm1.7512}$ |
| Greedy | $\mathbf{0.6943}_{\pm0.0732}$ | $\mathbf{2.8333}_{\pm1.6021}$ |
| Greedy (Merged) | $0.7537_{\pm0.2652}$ | $6.5833_{\pm4.3637}$ |
| CMAES | $1.2356_{\pm0.5295}$ | $10.3333_{\pm3.4448}$ |
| MA | $0.9067_{\pm0.1809}$ | $8.8333_{\pm2.4014}$ |
| NE-Stack | $0.7562_{\pm0.1836}$ | $5.3333_{\pm4.6762}$ |
| NE-Stacking (Merged) | $0.7428_{\pm0.2208}$ | $5.5833_{\pm3.3529}$ |
| NE-MA | $\underline{0.6952}_{\pm0.0730}$ | $\mathbf{2.8333}_{\pm2.2286}$ |
| NE-MA (Merged) | $0.7461_{\pm0.2084}$ | $6.2500_{\pm2.6410}$ |

(a)

(b)

Figure 14: Merged-split baselines on the *FTC* metadataset: (a) Test set performance with merged training, (b) Critical Difference diagram.

## L Neural Ensemblers Operating on the Original Input Space

In Section 3.1, we discussed that the Neural Ensembler in model-evaraging mode (**NE-MA**) computes weights $\theta_m(z;\beta)$ that rely solely on the base model predictions $z$ for each instance. Specifically, the weights are defined as:

$$\theta_m(z;\beta) = \frac{\exp f_m(z;\beta)}{\sum_{m'} f_{m'}(z;\beta)} \tag{22}$$

where $z$ represents the base model predictions for an instance $x \in \mathcal{X}$, and $\mathcal{X}$ denotes the original input space. As our experiments encompass different data modalities, $x$ can be a vector of tabular descriptors, an image, or text.

An alternative formulation involves computing the ensemble weights directly from the original input instances instead of using the base model predictions. This approach modifies the weight computation to:

$$\hat{y} = \sum_m \theta_m(x; \beta) \cdot z_m(x) = \sum_m \frac{\exp f_m(x; \beta)}{\sum_{m'} \exp f_{m'}(x; \beta)} \cdot z_m(x) \tag{23}$$

where $f_m$ now operates on the original input space $\mathcal{X}$ to produce unnormalized weights.

However, adopting this formulation introduces challenges in selecting an appropriate function $f_m$ for different data modalities. For example, if the instances are images, $f_m$ must be a network capable of processing images, such as a convolutional neural network. This requirement prevents us from using the same architecture across all modalities, limiting the generalizability of the approach.

To evaluate this idea, we tested this alternative neural ensembler on the *Scikit-learn Pipelines* metadataset, which consists of tabular data. We implemented $f_m$ as a four-layer MLP. Our results, represented by the yellow bar in Figure 13b, indicate that this strategy does not outperform the original approach proposed in Section 3, which uses the base model predictions as input.

We hypothesize that computing ensemble weights directly from the original input space may be more susceptible to overfitting, especially when dealing with datasets that have noisy or high-dimensional features. Additionally, this strategy may require tuning the hyperparameters of the network $f_m$ for each dataset to achieve optimal performance, reducing its effectiveness and generalizability across diverse datasets.

## M  Do Neural Ensemblers need a strong group of base models, i.e. found using Bayesian Optimization?

**Experimental Protocol**. Practitioners use some methods such as greedy ensembling as post-hoc ensemblers, i.e., they consider a set of models selected by a search algorithm such as Bayesian Optimization as base learners. *DivBO* enhances the Bayesian Optimization by accounting for the diversity in the ensemble in the acquisition function. We run experiments to understand whether the Neural Ensemblers' performance depends on a strong subset of 50 base models selected by *DivBO*, and whether it can help other methods. We conduct additional experiments by randomly selecting 50 models to understand the impact and significance of merely using a smaller set of base models. We normalize the base of the metric on the *single-best* base model from the complete set contained in the respective dataset.

**Results**. We report in Table 7 the results with the two selection methods (random and *DivBO*) using a subset of common baselines, where we normalize using the metric of the *single-best* from the whole set of models. We can compare directly with the results in Table 1. We observe that reducing the number of base models with *DivBO* negatively affects the performance of the Neural Ensemblers. Surprisingly, randomly selecting the subset of base models improves the results in two metadatasets (*TR-Class* and *NB-1000*). We hypothesize that decreasing the number of base models is beneficial for these metadatasets. With over 1000 base models available, the likelihood of identifying a preferred model and overfitting the validation data increases in these metadatasets. Naturally, decreasing the number of base models can also be detrimental for the Neural Ensemblers, as this happens for some metadatasets such as *TR-Reg* and *QT-Micro*. In contrast to the Neural Ensemblers, selecting a subset of strong models with *DivBO* improves the performance for some baselines such as Model Averaging (MA) or *TopK* ($K = 25$). In other words, it works as a preprocessing method for these ensembling approaches. Overall, the results in Table 7 demonstrate that **Neural Ensemblers do not need a strong group of base models to achieve competitive results**.

Table 7: Average NLL for Subset of Base Models selected Randomly or using DivBO

| | FTC | NB-Micro | NB-Mini | QT-Micro | QT-Mini | TR-Class | TR-Reg |
|---|---|---|---|---|---|---|---|
| **Single** | $1.0000_{\pm 0.0000}$ | $1.0000_{\pm 0.0000}$ | $1.0000_{\pm 0.0000}$ | $\mathit{1.0000}_{\pm 0.0000}$ | $1.0000_{\pm 0.0000}$ | $1.0000_{\pm 0.0000}$ | $\mathit{1.0000}_{\pm 0.0000}$ |
| **Single + DivBO** | $1.0000_{\pm 0.0000}$ | $1.0000_{\pm 0.0000}$ | $0.8707_{\pm 0.3094}$ | $1.7584_{\pm 2.0556}$ | $1.1846_{\pm 0.2507}$ | $1.1033_{\pm 0.9951}$ | $1.0039_{\pm 0.0424}$ |
| **Random + DivBO** | $0.9305_{\pm 0.3286}$ | $0.6538_{\pm 0.2123}$ | $0.9724_{\pm 0.0478}$ | $1.1962_{\pm 1.0189}$ | $0.9717_{\pm 0.1919}$ | $1.0107_{\pm 0.3431}$ | $1.0302_{\pm 0.1250}$ |
| **Top25 + DivBO** | $0.7617_{\pm 0.1136}$ | $\mathbf{0.5564}_{\pm 0.1961}$ | $0.9762_{\pm 0.0413}$ | $1.1631_{\pm 0.9823}$ | $0.9431_{\pm 0.2035}$ | $1.0023_{\pm 0.3411}$ | $1.0247_{\pm 0.1473}$ |
| **Quick + DivBO** | $0.7235_{\pm 0.0782}$ | $0.6137_{\pm 0.1945}$ | $0.9646_{\pm 0.0614}$ | $1.2427_{\pm 1.1130}$ | $0.9544_{\pm 0.2050}$ | $1.0014_{\pm 0.3423}$ | $1.0400_{\pm 0.1949}$ |
| **Greedy + DivBO** | $0.7024_{\pm 0.0720}$ | $0.6839_{\pm 0.3003}$ | $0.9762_{\pm 0.0413}$ | $1.1659_{\pm 0.9789}$ | $0.9435_{\pm 0.2029}$ | $1.0024_{\pm 0.3410}$ | $1.0271_{\pm 0.1531}$ |
| **CMAES + DivBO** | $0.8915_{\pm 0.1759}$ | $1.0000_{\pm 0.0000}$ | $1.0000_{\pm 0.0000}$ | $\mathit{1.0000}_{\pm 0.0000}$ | $1.0000_{\pm 0.0000}$ | $1.0166_{\pm 0.3319}$ | $1.0265_{\pm 0.1521}$ |
| **Random Forest + DivBO** | $0.7932_{\pm 0.1194}$ | $0.9338_{\pm 0.3436}$ | $1.1609_{\pm 0.2787}$ | $2.5998_{\pm 2.6475}$ | $1.7330_{\pm 0.9898}$ | $1.3308_{\pm 0.6640}$ | $1.0559_{\pm 0.1240}$ |
| **Gradient Boosting + DivBO** | $0.7908_{\pm 0.1848}$ | $1.4011_{\pm 0.5359}$ | $1.0000_{\pm 0.0000}$ | $3.1558_{\pm 1.8843}$ | $3.0103_{\pm 1.3714}$ | $1.3173_{\pm 1.1519}$ | $1.1285_{\pm 0.4153}$ |
| **Linear + DivBO** | $0.7433_{\pm 0.0870}$ | $0.6471_{\pm 0.2272}$ | $1.0199_{\pm 0.0345}$ | $1.5843_{\pm 1.6159}$ | $1.2392_{\pm 0.5343}$ | $1.0786_{\pm 1.0100}$ | $1.0326_{\pm 0.1265}$ |
| **SVM + DivBO** | $0.8312_{\pm 0.0943}$ | $0.7406_{\pm 0.2612}$ | $1.0730_{\pm 0.1264}$ | $5.5177_{\pm 3.3684}$ | $5.0079_{\pm 3.4926}$ | $1.4542_{\pm 1.4518}$ | $2.7702_{\pm 2.9398}$ |
| **XGB + DivBO** | $0.7884_{\pm 0.1286}$ | $0.7519_{\pm 0.2361}$ | $1.0000_{\pm 0.0000}$ | $3.1705_{\pm 2.8221}$ | $2.0159_{\pm 1.8026}$ | $1.7698_{\pm 1.6369}$ | $1.3226_{\pm 0.6368}$ |
| **CatBoost + DivBO** | $\mathit{0.6941}_{\pm 0.0953}$ | $0.8110_{\pm 0.2405}$ | $1.0000_{\pm 0.0000}$ | $2.3936_{\pm 2.4785}$ | $2.4792_{\pm 2.3078}$ | $1.3149_{\pm 1.3954}$ | $1.0730_{\pm 0.0815}$ |
| **LightGBM + DivBO** | $0.7706_{\pm 0.1919}$ | $3.7392_{\pm 2.7136}$ | $1.0000_{\pm 0.0000}$ | $2.1979_{\pm 2.2093}$ | $2.2706_{\pm 2.2088}$ | $1.6938_{\pm 1.1246}$ | $1.6427_{\pm 2.0133}$ |
| **Akaike + DivBO** | $0.8757_{\pm 0.0944}$ | $0.8159_{\pm 0.2196}$ | $1.0000_{\pm 0.0000}$ | $1.2270_{\pm 1.0049}$ | $1.0452_{\pm 0.2362}$ | $1.0202_{\pm 0.3379}$ | $1.0564_{\pm 0.1884}$ |
| **MA + DivBO** | $0.7245_{\pm 0.0788}$ | $0.5712_{\pm 0.2185}$ | $0.9678_{\pm 0.0558}$ | $1.0559_{\pm 0.7452}$ | $0.9501_{\pm 0.1617}$ | $1.0068_{\pm 0.4141}$ | $1.0237_{\pm 0.1502}$ |
| **Single + Random** | $1.0067_{\pm 0.0164}$ | $1.0000_{\pm 0.0000}$ | $0.9240_{\pm 0.3504}$ | $1.2915_{\pm 0.9952}$ | $1.1261_{\pm 0.3134}$ | $1.0225_{\pm 0.3353}$ | $1.1378_{\pm 0.4641}$ |
| **Top25 + Random** | $0.8397_{\pm 0.1000}$ | $0.5848_{\pm 0.1980}$ | $\mathit{0.6526}_{\pm 0.3019}$ | $3.6553_{\pm 2.7053}$ | $3.0436_{\pm 2.1378}$ | $1.2599_{\pm 1.5015}$ | $1.0611_{\pm 0.2799}$ |
| **Quick + Random** | $0.7305_{\pm 0.0764}$ | $0.5958_{\pm 0.1917}$ | $0.6656_{\pm 0.2968}$ | $1.7769_{\pm 2.1443}$ | $1.1646_{\pm 0.3728}$ | $1.0797_{\pm 1.0007}$ | $1.0151_{\pm 0.1546}$ |
| **Greedy + Random** | $0.7024_{\pm 0.0720}$ | $0.5783_{\pm 0.1857}$ | $0.6617_{\pm 0.2839}$ | $1.6723_{\pm 2.1446}$ | $0.9961_{\pm 0.1290}$ | $1.0725_{\pm 0.9978}$ | $1.0023_{\pm 0.0961}$ |
| **CMAES + Random** | $1.2164_{\pm 0.3851}$ | $1.0000_{\pm 0.0000}$ | $1.0000_{\pm 0.0000}$ | $\mathit{1.0000}_{\pm 0.0000}$ | $1.0000_{\pm 0.0000}$ | $1.1579_{\pm 0.9913}$ | $1.0004_{\pm 0.0851}$ |
| **RF + Random** | $0.7505_{\pm 0.0915}$ | $0.8325_{\pm 0.2255}$ | $0.8654_{\pm 0.3179}$ | $3.8684_{\pm 2.8060}$ | $3.0156_{\pm 1.9975}$ | $1.2034_{\pm 0.4063}$ | $1.0079_{\pm 0.0795}$ |
| **GBT + Random** | $0.7235_{\pm 0.1605}$ | $1.8377_{\pm 1.4510}$ | $0.9533_{\pm 0.0809}$ | $2.3407_{\pm 1.3498}$ | $3.1659_{\pm 2.0795}$ | $1.3344_{\pm 1.1611}$ | $1.0500_{\pm 0.1940}$ |
| **Linear + Random** | $0.7541_{\pm 0.0897}$ | $0.7400_{\pm 0.2827}$ | $0.7732_{\pm 0.2331}$ | $2.0502_{\pm 2.2932}$ | $1.9028_{\pm 1.3239}$ | $1.0423_{\pm 1.0174}$ | $1.0430_{\pm 0.1224}$ |
| **SVM + Random** | $0.8010_{\pm 0.0903}$ | $0.7767_{\pm 0.3006}$ | $0.9268_{\pm 0.5498}$ | $5.4773_{\pm 3.3607}$ | $5.5665_{\pm 3.3068}$ | $1.3964_{\pm 1.4267}$ | $2.8271_{\pm 3.0203}$ |
| **XGB + Random** | $0.8288_{\pm 0.1463}$ | $0.7369_{\pm 0.2384}$ | $0.8875_{\pm 0.5032}$ | $3.7747_{\pm 3.1635}$ | $2.6163_{\pm 2.1103}$ | $1.7214_{\pm 1.5419}$ | $1.1962_{\pm 0.3298}$ |
| **CatBoost + Random** | $\mathbf{0.6911}_{\pm 0.1007}$ | $0.8294_{\pm 0.2366}$ | $0.9472_{\pm 0.4823}$ | $2.6695_{\pm 2.6137}$ | $2.7315_{\pm 2.3219}$ | $1.2006_{\pm 1.2116}$ | $1.0559_{\pm 0.2144}$ |
| **LightGBM + Random** | $0.7960_{\pm 0.1977}$ | $3.6975_{\pm 2.6631}$ | $5.2689_{\pm 4.6347}$ | $2.8698_{\pm 2.6137}$ | $3.6272_{\pm 3.2919}$ | $1.7133_{\pm 1.0934}$ | $1.6848_{\pm 2.1630}$ |
| **Akaike + Random** | $0.8045_{\pm 0.0987}$ | $0.6538_{\pm 0.1831}$ | $0.6905_{\pm 0.2981}$ | $2.2022_{\pm 2.4118}$ | $1.5073_{\pm 0.6848}$ | $1.1180_{\pm 1.0381}$ | $\mathbf{0.9970}_{\pm 0.0886}$ |
| **MA + Random** | $0.9069_{\pm 0.1812}$ | $0.8677_{\pm 0.2292}$ | $0.6698_{\pm 0.2898}$ | $4.8593_{\pm 3.1360}$ | $3.4575_{\pm 2.6490}$ | $1.4759_{\pm 1.9396}$ | $1.4286_{\pm 1.7242}$ |
| **NE(S) + Random** | $0.7709_{\pm 0.2204}$ | $0.7551_{\pm 0.2493}$ | $\mathbf{0.6187}_{\pm 0.2950}$ | $\mathbf{0.8292}_{\pm 0.5466}$ | $\mathbf{0.8160}_{\pm 0.3852}$ | $\mathbf{0.9540}_{\pm 0.5077}$ | $4.2183_{\pm 3.4808}$ |
| **NE(MA) + Random** | $0.6972_{\pm 0.0712}$ | $0.7911_{\pm 0.2147}$ | $0.6650_{\pm 0.2750}$ | $1.6877_{\pm 2.1535}$ | $1.0903_{\pm 0.2578}$ | $1.0674_{\pm 0.9998}$ | $1.0277_{\pm 0.1994}$ |
| **NE(S) + DivBO** | $0.7715_{\pm 0.2141}$ | $0.6204_{\pm 0.2234}$ | $1.0000_{\pm 0.0000}$ | $1.5040_{\pm 1.9442}$ | $\mathit{0.8329}_{\pm 0.2659}$ | $\mathit{0.9729}_{\pm 0.3952}$ | $6.9453_{\pm 3.4749}$ |
| **NE(MA) + DivBO** | $0.7036_{\pm 0.0698}$ | $\mathit{0.5704}_{\pm 0.2345}$ | $1.0000_{\pm 0.0000}$ | $1.1237_{\pm 0.9964}$ | $0.9200_{\pm 0.1966}$ | $1.0016_{\pm 0.3407}$ | $1.0070_{\pm 0.0977}$ |

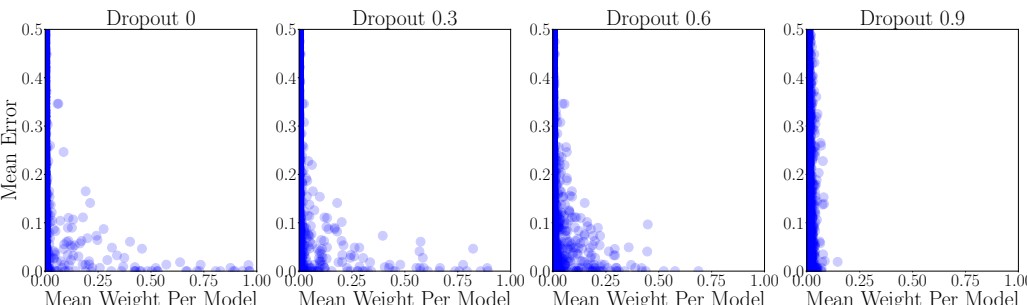

Figure 15: Mean weights assigned to the base models decrease with DropOut rate. Every data point is a model. The errors and weights are the mean values across many datasets for every model.

## N Visualizing the effect of the regularization on the model weights

In Figure 15, we study the effect of the DropOut rate on the ensembler weights, in Model-Averaging model using the *QT-Micro* meta-dataset. When there is no DropOut some weights are close to one, i.e. they are preferred models. As we increase the value, many models with high weights decrease. If the rate is very high (e.g. 0.9), we will have many models contributing to the ensemble, with weights different from zero.

# O  Details On Meta-Datasets

In Table 11 we provide further information about the meta-datasets, such as the number of datasets, the average samples in the test and validation splits and the average number of classes.

## O.1  Scikit learn Pipelines

**Search Space.** Our primary motivation is to investigate the ensembling of automated machine learning pipelines to enhance performance across various classification tasks. To effectively study ensembling methods and benchmark different strategies, we require a diverse set of pipelines. Therefore, we construct a comprehensive search space inspired by the TPOT library (Olson et al., 2016), encompassing a wide range of preprocessors, feature selectors, and classifiers. The pipelines are structured in three stages: preprocessor, feature selector, and classifier. This allows us to explore numerous configurations systematically. This extensive and diverse search space enables us to examine the impact of ensembling on a variety of models and serves as a robust benchmark for evaluating different ensembling techniques. Detailed descriptions of the components and their hyperparameters are provided in Tables 8, 9, and 10.

**Datasets** We utilized the OpenML Curated Classification benchmark suite 2018 (OpenML-CC18) (Bischl et al., 2019) as the foundation for our meta-dataset. OpenML-CC18 comprises 72 diverse classification datasets carefully selected to represent a wide spectrum of real-world problems, varying in size, dimensionality, number of classes, and domains. This selection ensures comprehensive coverage across various types of classification tasks, providing a robust platform for evaluating the performance and generalizability of different ensembling approaches.

**Meta-Dataset Creation** To construct our meta-dataset, we randomly selected 500 pipeline configurations for each dataset from our comprehensive search space. Each pipeline execution was constrained to a maximum runtime of 15 minutes. During this process, we had to exclude three datasets (*connect-4, Devnagari-Script, Internet-Advertisements*) due to excessive computational demands that exceeded our runtime constraints. For data preprocessing, we standardized the datasets by removing missing values and encoding categorical features. We intentionally left other preprocessing tasks to be handled autonomously by the pipelines themselves, allowing them to adapt to the specific characteristics of each dataset. This approach ensures that the pipelines can perform necessary transformations such as scaling, normalization, or feature engineering based on their internal configurations, which aligns with our objective of evaluating automated machine learning pipelines in a realistic setting.

Table 8: Classifiers and their hyperparameters used in the TPOT search space.

| Classifier | Hyperparameters |
|---|---|
| sklearn.naive_bayes.GaussianNB | None |
| sklearn.naive_bayes.BernoulliNB | alpha (float, [1e-3, 100.0], default=50.0) |
| | fit_prior (categorical, {True, False}) |
| sklearn.naive_bayes.MultinomialNB | alpha (float, [1e-3, 100.0], default=50.0) |
| | fit_prior (categorical, {True, False}) |
| sklearn.tree.DecisionTreeClassifier | criterion (categorical, {'gini', 'entropy'}) |
| | max_depth (int, [1, 10], default=5) |
| | min_samples_split (int, [2, 20], default=11) |
| | min_samples_leaf (int, [1, 20], default=11) |
| sklearn.ensemble.ExtraTreesClassifier | n_estimators (constant, 100) |
| | criterion (categorical, {'gini', 'entropy'}) |
| | max_features (float, [0.05, 1.0], default=0.525) |
| | min_samples_split (int, [2, 20], default=11) |
| | min_samples_leaf (int, [1, 20], default=11) |
| | bootstrap (categorical, {True, False}) |
| sklearn.ensemble.RandomForestClassifier | n_estimators (constant, 100) |
| | criterion (categorical, {'gini', 'entropy'}) |
| | max_features (float, [0.05, 1.0], default=0.525) |
| | min_samples_split (int, [2, 20], default=11) |
| | min_samples_leaf (int, [1, 20], default=11) |
| | bootstrap (categorical, {True, False}) |
| sklearn.ensemble.GradientBoostingClassifier | n_estimators (constant, 100) |
| | learning_rate (float, [1e-3, 1.0], default=0.5) |
| | max_depth (int, [1, 10], default=5) |
| | min_samples_split (int, [2, 20], default=11) |
| | min_samples_leaf (int, [1, 20], default=11) |
| | subsample (float, [0.05, 1.0], default=0.525) |
| | max_features (float, [0.05, 1.0], default=0.525) |
| sklearn.neighbors.KNeighborsClassifier | n_neighbors (int, [1, 100], default=50) |
| | weights (categorical, {'uniform', 'distance'}) |
| | p (categorical, {1, 2}) |
| sklearn.linear_model.LogisticRegression | penalty (categorical, {'l1', 'l2'}) |
| | C (float, [1e-4, 25.0], default=12.525) |
| | dual (categorical, {True, False}) |
| | solver (constant, 'liblinear') |
| xgboost.XGBClassifier | n_estimators (constant, 100) |
| | max_depth (int, [1, 10], default=5) |
| | learning_rate (float, [1e-3, 1.0], default=0.5) |
| | subsample (float, [0.05, 1.0], default=0.525) |
| | min_child_weight (int, [1, 20], default=11) |
| | n_jobs (constant, 1) |
| | verbosity (constant, 0) |
| sklearn.linear_model.SGDClassifier | loss (categorical, {'log_loss', 'modified_huber'}) |
| | penalty (categorical, {'elasticnet'}) |
| | alpha (float, [0.0, 0.01], default=0.005) |
| | learning_rate (categorical, {'invscaling', 'constant'}) |
| | fit_intercept (categorical, {True, False}) |
| | l1_ratio (float, [0.0, 1.0], default=0.5) |
| | eta0 (float, [0.01, 1.0], default=0.505) |
| | power_t (float, [0.0, 100.0], default=50.0) |
| sklearn.neural_network.MLPClassifier | alpha (float, [1e-4, 0.1], default=0.05) |
| | learning_rate_init (float, [0.0, 1.0], default=0.5) |

Table 9: Preprocessors and their hyperparameters used in the TPOT search space.

| Preprocessor | Hyperparameters |
|---|---|
| None | None |
| sklearn.preprocessing.Binarizer | threshold (float, [0.0, 1.0], default=0.5) |
| sklearn.decomposition.FastICA | tol (float, [0.0, 1.0], default=0.0) |
| sklearn.cluster.FeatureAgglomeration | linkage (categorical, {'ward', 'complete', 'average'}) |
| | metric (categorical, {'euclidean', 'l1', 'l2', 'manhattan', 'cosine'}) |
| sklearn.preprocessing.MaxAbsScaler | None |
| sklearn.preprocessing.MinMaxScaler | None |
| sklearn.preprocessing.Normalizer | norm (categorical, {'l1', 'l2', 'max'}) |
| sklearn.kernel_approximation.Nystroem | kernel (categorical, {'rbf', 'cosine', 'chi2', 'laplacian', 'polynomial', 'poly', 'linear', 'additive_chi2', 'sigmoid'}) |
| | gamma (float, [0.0, 1.0], default=0.5) |
| | n_components (int, [1, 10], default=5) |
| sklearn.decomposition.PCA | svd_solver (categorical, {'randomized'}) |
| | iterated_power (int, [1, 10], default=5) |
| sklearn.preprocessing.PolynomialFeatures | degree (constant, 2) |
| | include_bias (categorical, {False}) |
| | interaction_only (categorical, {False}) |
| sklearn.kernel_approximation.RBFSampler | gamma (float, [0.0, 1.0], default=0.5) |
| sklearn.preprocessing.RobustScaler | None |
| sklearn.preprocessing.StandardScaler | None |
| tpot.builtins.ZeroCount | None |
| tpot.builtins.OneHotEncoder | minimum_fraction (float, [0.05, 0.25], default=0.15) |
| | sparse (categorical, {False}) |
| | threshold (constant, 10) |

Table 10: Feature selectors and their hyperparameters used in the TPOT search space.

| Selector | Hyperparameters |
|---|---|
| None | None |
| sklearn.feature_selection.SelectFwe | alpha (float, [0.0, 0.05], default=0.025) |
| sklearn.feature_selection.SelectPercentile | percentile (int, [1, 100], default=50) |
| sklearn.feature_selection.VarianceThreshold | threshold (float, [0.0001, 0.2], default=0.1) |
| sklearn.feature_selection.RFE | step (float, [0.05, 1.0], default=0.525) |
| | estimator (categorical, {'sklearn.ensemble.ExtraTreesClassifier'}) |
| | **Estimator Hyperparameters:** |
| |   n_estimators (constant, 100) |
| |   criterion (categorical, {'gini', 'entropy'}) |
| |   max_features (float, [0.05, 1.0], default=0.525) |
| sklearn.feature_selection.SelectFromModel | threshold (float, [0.0, 1.0], default=0.5) |
| | estimator (categorical, {'sklearn.ensemble.ExtraTreesClassifier'}) |
| | **Estimator Hyperparameters:** |
| |   n_estimators (constant, 100) |
| |   criterion (categorical, {'gini', 'entropy'}) |
| |   max_features (float, [0.05, 1.0], default=0.525) |

Table 11: Metadatasets Information

| Meta-Dataset | Modality | Task Information | No. Datasets | Avg. Samples for Validation | Avg. Samples for Test | Avg. Models per Dataset | Avg. Classes per Dataset |
|---|---|---|---|---|---|---|---|
| Nasbench (100) | Vision | NAS, Classification (Dong and Yang, 2020) | 3 | 11000 | 6000 | 100 | 76.6 |
| Nasbench (1K) | Vision | NAS, Classification (Dong and Yang, 2020) | 3 | 11000 | 6000 | 1K | 76.6 |
| QuickTune (Micro) | Vision | Finetuning, Classification (Arango et al., 2024a) | 30 | 160 | 160 | 255 | 20. |
| QuickTune (Mini) | Vision | Finetuning, Classification (Arango et al., 2024a) | 30 | 1088 | 1088 | 203 | 136. |
| FTC | Language | Finetuning, Classification, Section (Arango et al., 2024b) | 6 | 39751 | 29957 | 105 | 4.6 |
| TabRepo Clas. | Tabular | Classification (Salinas and Erickson, 2023) | 83 | 1134 | 126 | 1530 | 3.4 |
| TabRepo Reg. | Tabular | Regression (Salinas and Erickson, 2023) | 17 | 3054 | 3397 | 1530 | - |
| Sk-Learn Pipelines. | Tabular | Classification, Section O | 69 | 1514 | 1514 | 500 | 5.08 |

