# OpenReview forum: "Regularized Neural Ensemblers"
_automl.cc/AutoML/2025/Methods_Track — AutoML 2025 Methods Track_

### Official Review · Reviewer_CgAA · 2025-04-29

**Comments To Authors:**

## Summary

The paper presents two approaches to creating ensembling regression or classification models. Two modes of models are proposed: stacking and model averaging.  To ensure the diversity of the models, regularization through model dropout was introduced. An empirical evaluation of the model was carried out.

## Potential impact on the field of AutoML

In particular, a new method for creating ensembling in **model averaging,** which assigns weights to model predictions for each observation separately, has been proposed. This is a new method for training dynamic ensembling.

It has the potential to have a major impact on AutoML as a new post-processing approach that often yields better results than optimizing individual models, plus it is often computationally cheaper.

### Strengths

- interesting is the **model averaging mode,** which creates dynamic weights for each model for each observation
- Used empirical evaluation on a broad set of 240 datasets
- Large set of comparison baselines based on ensemblings of different models (e.g. Greedy, CMAES), but stacking with different aggregation models e.g. Random Forest or CatBoost

### Weaknesses

- I do not see how mode **stacking** differs from previously defined stacking with MLP models as the final estimating model, which is, for example, implemented in Autogluon?
- What set of models were available for each ensemblin
- Instead of normalizing in this way, I would propose the use of normalization analogous to ADTM (Average Distance to Minimum/Maximum)
- No results in the tables for KNOP, KNORAE, MetaDES methods
- The main text does not include graphs showing the lack of statistical significance occurring between most models for aggregated data (Figure 7) - this is difficult to pick out from the large Tables 1 and 2

### Minor comments

- Editorially, the paper is underdeveloped. Descriptions of tables and graphs (e.g., Figure 3) are not informative. The abbreviation NLL is not described.
- A great deal of results are in the appendix which makes the paper much more difficult to read.
- The appendix is interesting on the one hand, but its size makes it impossible to analyze the results in detail, so I do not undertake a thorough evaluation of it.
- an ablation study was conducted (in the appendix)

**Review Confidence:**

4

**Review Rating:**

4

---

### Official Review · Reviewer_fbb7 · 2025-05-04

**Comments To Authors:**

**Summary**
This paper proposes an ensemble method that is capable of combining the predictions of many models using input-dependent adaptive reweighting guided by a neural network. Additionally, the authors propose a regularization technique that aims to improve the diversity of outputs from the models in the ensemble, which the authors justify theoretically. The authors then validate their method across a diverse set of tabular, vision, and language tasks.

**Strengths**
- The motivating experiment (shown in Figure 1) showing the need for dynamic ensembles is convincing, and is a nice touch.
- The theoretical justification for the regularization, which lower bounds diversity, is useful.
- The experimental results seem quite thorough. I find the NAS-Bench experiment to be particularly interesting.

**Weaknesses/questions**
- It would be useful to see how different choices of regularization affect performance, beyond tuning the dropout rate. Is dropout the best option to improve diversity, or can you also directly add the diversity measure to the objective function?
- How does the ensemble compare to dynamically selecting the argmax model from the predicted ensemble weights, per example? This approach would then resemble routing, and would open up an interesting discussion about the relationship between these types of methods.

**Other comments**
- Possible typo on line 80, "p(m|m)" should maybe be p(m|x)?
- Possible typo on 112. Missing exp in the denominator?

**Review Confidence:**

3

**Review Rating:**

8

---

### Official Review · Reviewer_eNk1 · 2025-05-13

**Comments To Authors:**

The paper proposes an instance-specific post-hoc ensembling method with a regularized neural work. Given an ensemble of (already trained) base models, the ensembling network receives as input the predictions from the base models and outputs either the predicted output (stacking mode) or the weights for combining the base models’ prediction (model-averaging mode). The ensembling network is regularized by randomly dropping some of the base models to avoid overfitting.

The paper tackles an important topic. Ensembling is widely adopted in ML, and weighting the base models dynamically can be beneficial compared to static weighting. The idea of a regularized ensembling neural network that receives the predictions (instead of the input features) as input is interesting and seems novel.

The paper is very well-written and clear. Experimental results across different domains (CV, NLP and tabular data) show good overall performance when compared to baselines. Ablation study is provided to confirm the positive impact of the regularization component. The additional research questions shown in the appendices also answer most of the questions I had in mind while reading the main text, such as what happen if we allow the base models to be trained on the combined dataset of training and validation.

Overall, I think this is a very nice work and would recommend acceptance.

Some minor issues:

- I understand that it can be difficult to fit all content within the page limit, but I think some of the bits that are in the appendices should be in the main text instead (e.g., the mentioning of mixture-of-experts in related work).

- Reproducibility: I browsed the provided anonymous github repo, and found that some of the files were not accessible (e.g., SearchingOptimalEnsembles/posthoc/greedy_ensembler.py)

**Review Confidence:**

3

**Review Rating:**

9

---

### Meta-Review · Area_Chair_L1tR · 2025-05-13

**Recommendation:** Accept
**Confidence:** 4

**Metareview:**

This paper proposes a method for performing ensembling via neural networks. The ensembling model is regularized via dropout of the input base model predictions. The benefits of the proposed approach are justified both theoretically and empirically.

The key strengths of this paper lie in its significance, novelty, and experimental results. The proposed post-processing method yields better results while remaining computational efficient. This potential impact is multiplied by the widespread use of ensembling in practice. The proposed approach is novel since it dynamically assigns weights to individual model predictions. The experimental validation is thorough, consisting of over 240 datasets and a large set of baselines. The results and ablation demonstrates the effectiveness of the proposed approach.

Areas of improvement center mainly on clarity. Reviewers raised the issue of interesting results and discussion being contained in the appendix that may be better suited in the main body of the paper. Captions for tables and figures can be improved, and select results may be better communicated as figures rather than tables. There are also minor concerns about reproducibility and relationship to related work.

Overall, the potential impact of this paper is a strong argument for its acceptance. This is supported by the novelty and experimental results. The weaknesses of the paper focused on clarity can likely be resolved during the camera-ready period, for example by incorporating some key parts of the appendix into the main paper. I recommend acceptance.